# Long non-coding RNA *DARS-AS1* promotes tumor progression by directly suppressing PACT-mediated cellular stress

Liuqing Yang[1,7], Kequan Lin[2,7], Lin Zhu[1], Huili Wang[3], Shuaishuai Teng[3], Lijun Huang[4], Shiyi Zhou[4], Guanbin Zhang ORCID [5,6], Zhi John Lu ORCID [1] & Dong Wang ORCID [4✉]

Cancer cells evolve various mechanisms to overcome cellular stresses and maintain progression. Protein kinase R (PKR) and its protein activator (PACT) are the initial responders in monitoring diverse stress signals and lead to inhibition of cell proliferation and cell apoptosis in consequence. However, the regulation of PACT-PKR pathway in cancer cells remains largely unknown. Herein, we identify that the long non-coding RNA (lncRNA) *aspartyl-tRNA synthetase antisense RNA 1* (*DARS-AS1*) is directly involved in the inhibition of the PACT-PKR pathway and promotes the proliferation of cancer cells. Using large-scale CRISPRi functional screening of 971 cancer-associated lncRNAs, we find that *DARS-AS1* is associated with significantly enhanced proliferation of cancer cells. Accordingly, knocking down *DARS-AS1* inhibits cell proliferation of multiple cancer cell lines and promotes cancer cell apoptosis in vitro and significantly reduces tumor growth in vivo. Mechanistically, *DARS-AS1* directly binds to the activator domain of PACT and prevents PACT-PKR interaction, thereby decreasing PKR activation, eIF2α phosphorylation and inhibiting apoptotic cell death. Clinically, *DARS-AS1* is broadly expressed across multiple cancers and the increased expression of this lncRNA indicates poor prognosis. This study elucidates the lncRNA *DARS-AS1* directed cancer-specific modulation of the PACT-PKR pathway and provides another target for cancer prognosis and therapeutic treatment.

[1] School of Life Science, Tsinghua University, 100084 Beijing, China. [2] High Performance Computing Department, National Supercomputing Center in Shenzhen, Shenzhen 518055 Guangdong, China. [3] School of Medicine, Tsinghua University, 100084 Beijing, China. [4] State Key Laboratory of Southwestern Chinese Medicine Resources, School of Basic Medical Sciences, Chengdu University of Traditional Chinese Medicine, Chengdu 611137, China. [5] Mianyang People's Hospital, Mianyang 621000, China. [6] National Engineering Research Center for Beijing Biochip Technology, 102206 Beijing, China. [7] These authors contributed equally: Liuqing Yang, Kequan Lin. ✉email: dwang@cdutcm.edu.cn

The ability to adapt to stresses is an essential feature for the survival and proliferation of cancer cell[1–4]. The rapid proliferation and metabolism characteristic of carcinomas culminate in a severe microenvironment—nutrient starvation, hypoxia, and low pH—which can trigger cell death signaling pathways. The dysregulation of stress response genes—such as p53[5], heat shock proteins[6,7], KRAS[8,9] and HIF-1[10–13]—are often observed in cancers, blocking apoptosis and promoting survival.

Protein kinase R (PKR) is an important stress sensor and a kinase of eukaryotic initiation factor 2α subunit (eIF2α), which is a translational regulator that connects cellular stresses with cell death. PKR was initially identified as an antiviral protein through sensing non-self double-strand RNAs (dsRNAs). When activated, PKR phosphorylates eIF2α to inhibit viral and cellular protein synthesis[14–16]. PACT (PKR-activating protein) has been identified as the first protein activator of PKR in the absence of dsRNA[17–23]. Through direct interaction with PKR, PACT transduces diverse stresses—serum starvation, and peroxide or arsenite treatment—to PKR and downstream signaling pathways. In addition to the phosphorylation of eIF2α, PACT-mediated PKR activation triggers various events involved in stress response, including altering redox status via the PI3K/Akt[24] pathway, enhancing DNA damage checking by p53[25,26], and regulating transcription through NF-κB[27–29]. Considering their critical roles in stress response, proliferation, apoptosis, and other crucial cellular processes, PKR and PACT are promising therapeutic targets in a plethora of diseases, most notably cancer[30–33]. However, despite this pleiotropic functionality and biological significance, the regulation of PACT/PKR activity in cancer cells remains elusive.

lncRNAs are transcripts greater than 200 nucleotides without protein-coding potential[34]. Since thousands of lncRNAs have been identified by advanced genome-wide sequencing projects[35,36], numerous efforts have been made to elucidate their biological functions. Accumulating studies demonstrate that lncRNAs are involved in a multitude of biological processes[37], including the regulation of X chromosome inactivation[38,39], imprinting[40], transcription[41,42], translation[43], and even cancer growth[44–47]. In these investigations, a number of lncRNAs have been reported to be involved in the PACT/PKR pathway. One such study demonstrated that lncRNA ASPACT inhibits PACT transcription and increases nuclear retention of PACT mRNA[48]. Other investigations have shown that lncRNA nc886 binds to PKR and represses its phosphorylation[49,50]. As yet, no lncRNA has been reported to regulate PACT-mediated PKR activation.

Aspartyl-tRNA synthetase antisense RNA 1 (DARS-AS1) has been identified as an oncogenic lncRNA[51–54]. Through the modulation of miR-194-5p[53], miR-129[52], and miR-532-3p[51], DARS-AS1 was shown to promote the growth of clear cell renal cancer, thyroid cancer, and non-small cell lung cancer, respectively. Tong and fellows also found that DARS-AS1 facilitates myeloma progression by maintaining RNA-binding motif protein 39 (RBM39) stability[55]. However, there has not been any investigation into whether this lncRNA is involved in the regulation of PACT-PKR activation as well as the stress responses of cancer cells.

Herein, we performed large-scale loss-of-function screening via a CRISPRi system and determined that lncRNA DARS-AS1 promotes the proliferation of multiple cancer cell types. Furthermore, we identified the underlying mechanism: DARS-AS1 directly binds to PACT, suppressing the association of PACT and PKR, preventing the phosphorylation of the PKR downstream substrate eIF2α and ultimately inhibiting apoptotic cell death. Taken together, our work reveals lncRNA DARS-AS1 as a regulator of the PACT-PKR pathway and a potential target for cancer therapy and prognosis.

## Results

**Large-scale CRISPRi screening identifies DARS-AS1 promotes colorectal cancer proliferation.** Extensive genome profiling studies have identified hundreds of lncRNAs associated with cancer. However, their functions remain largely unknown[56]. To pinpoint promising lncRNA candidates involved in cancer progression, we performed loss-of-function screening for reduced proliferation in the colorectal cancer cell line SW620 using the CRISPRi system (Fig. 1a). A unique feature of SW480 and SW620 colon carcinoma cell lines is that they are derived from the primary and secondary tumors of a single patient. This enables a valuable comparison for examining the genetic changes in late colon cancer progression[57]. Accordingly, we profiled the transcriptome of colorectal cancer cell lines (SW480 and SW620) using RNA-sequencing and also collected some potential functional lncRNAs from published literatures. From these results, we designed a pooled sgRNA library, containing 7355 sgRNA oligos targeting 971 cancer-related lncRNAs and 500 non-targeting sgRNA oligos for negative controls (Supplementary Data 1).

Following plasmids construction and lentiviral packaging, we transduced the dCas9-SW620 colorectal cancer cell line with the aforementioned library in four independent infection experiments. The multiplicity of infection (MOI) of these infections was 0.1–0.3, indicating that each cell can be transfected by only one sgRNA. After in vitro culture for 18 days, the distribution of enrichment of the targeting sgRNAs had reduced or increased post-screening while the abundance of non-targeting control oligos remained relatively unchanged from the pre-screening distribution, indicating a high screening specificity for our targeting library (Fig. 1b and Supplementary Table 1). LINC00205, which was previously reported to promote lung cancer and liver cancer progression[58–60], was screened out (log2(foldchange) < −0.58, p value < 0.05), confirming the reliability of this screening (Fig. 1b).

Among all lncRNAs tested, DARS-AS1 was also screened out with three of its associated sgRNA oligos significantly reduced after 18 days culture, indicating that the knockdown of this lncRNA results in reduced cancer proliferation (Fig. 1b). This result was further validated by MTS assay in colorectal cancer cells, which showed that the growth rate of DARS-AS1-knockdown cells reduced to only half of the control cells (Fig. 1c), and is consistent with previous reports on several other cancer types: clear cell renal cancer, thyroid cancer and non-small cell lung cancer[51–53,55]. However, its function and molecular mechanisms in colorectal cancer remain uninvestigated. Therefore, we selected this lncRNA for further study.

**DARS-AS1 is highly expressed in multiple cancers.** To examine the expression of DARS-AS1 in patients, we comprehensively analyzed 10327 tumor samples from the Cancer Genome Atlas (TCGA) project. Our results show that DARS-AS1 is broadly expressed and significantly upregulated over healthy cells in a variety of tumors, including colon adenocarcinoma (COAD), kidney renal clear cell carcinoma (KIRC), and kidney renal papillary cell carcinoma (KIRP) to name a few (Fig. 1d and Supplementary Fig. 1a, b). Analysis of paired healthy/tumor samples further confirmed a significantly higher expression of DARS-AS1 in the tumors of bladder urothelial carcinoma (BLCA), kidney renal clear cell carcinoma (KIRC), prostate adenocarcinoma (PRAD), lung squamous cell carcinoma (LUSC), uterine corpus endometrial carcinoma (UCEC), lung adenocarcinoma (LUAD), liver hepatocellular carcinoma (LIHC), kidney renal papillary cell carcinoma (KIRP), and colon adenocarcinoma (COAD) (p value < 0.05) (Fig. 1e–m).

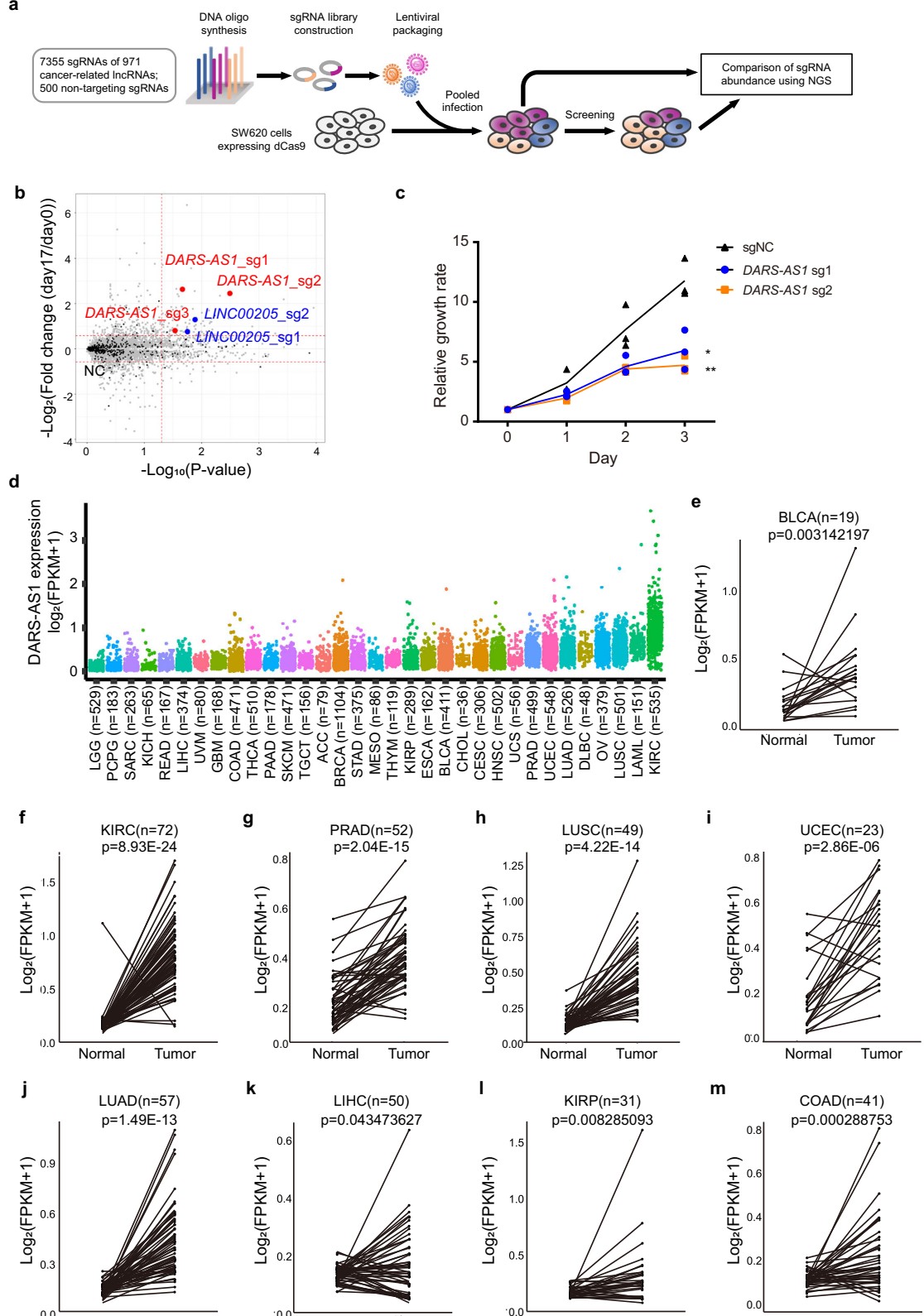

**Fig. 1 *DARS-AS1* is characterized as an oncogenic lncRNA. a** Schematic representation of screening using the CRISPRi system. **b** Enrichment of sgRNAs after screening. The horizontal dotted lines represent log₂(Fold change) = ±0.58. The vertical dotted line represents *p* value = 0.05. Black dots are non-targeting sgRNAs (marked as NC). Red dots are sgRNAs targeting *DARS-AS1*. Blue dots are sgRNAs targeting *LINC00205*, a previously reported oncogenic lncRNA. Fold change = (normalized reads day 17)/(normalized reads day 0). **c** Knockdown *DARS-AS1* by sgRNAs inhibits cell growth. Error bars represent ±SD in triplicate experiments. \**p* ≤ 0.05, \*\**p* ≤ 0.01, by two-tailed Student's *t* test. **d** Expression of *DARS-AS1* in tumors (TCGA datasets). **e**–**m** Expression of *DARS-AS1* in paired normal and tumor samples from BLCA, KIRC, PRAD, LUSC, UCEC, LUAD, LIHC, KIRP and COAD patients, respectively (TCGA datasets). *p* values were obtained by paired two-tailed Student's *t* test.

Collectively, these results demonstrate that *DARS-AS1* is broadly and highly expressed across a multitude of carcinomas.

**Knockdown of *DARS-AS1* suppresses cell proliferation in multiple cancers**. Since *DARS-AS1* and *DARS*—the antisense-stranded coding gene—share the same promoter and are adjacent to one another, we designed shRNAs to specifically knockdown *DARS-AS1* but not *DARS* (Supplementary Fig. 2a, b and Supplementary Table 2). Beside SW620, we also used three other cell lines with higher expression level of *DARS-AS1* to detect the knockdown efficiency and function of shRNAs (Supplementary Table 3). Our results show that all three designed shRNAs achieve at least an 80% knockdown efficiency of *DARS-AS1*, while have little impact on the abundance of DARS mRNA (Supplementary Fig. 2c–f). Moreover, we found that the knockdown of *DARS-AS1* by these shRNAs significantly inhibit cell growth of colorectal cancer cell lines SW620 (by 49.7%) and HCT116 (by 27.7%), breast cancer cell line MBA-MD-231 (by 53.4%) and liver cancer cell line HepG2 (by 92.7%), as well as their ability of unanchored sphere formation (reduced about 50.8%, 44.6%, 40.7% and 75.7% on average for each cell line) (Fig. 2a, b). The results of colony formation assay in SW620 further validated that cell proliferation is dramatically repressed by *DARS-AS1* shRNAs, reduced about 69.6% on average (Fig. 2c).

To complement the loss-of-function studies, we next generated *DARS-AS1*-overexpressing SW620 cells (Supplementary Fig. 2g). Overexpression *DARS-AS1* significantly increases cell growth (by 1.8-fold), unanchored sphere formation (by 1.4-fold) and colony formation (by 3.3-fold) of SW620 cells (Fig. 2d–f). We used another lower *DARS-AS1*-expressed cell line, A549, to confirm this result. This cell proliferation enhancement by overexpression of *DARS-AS1* was further observed in A549 cells (Supplementary Fig. 2h, i and Supplementary Table 3). Together, these gain- and loss-of-function studies demonstrate that *DARS-AS1* promotes cancer cell proliferation in vitro.

**_DARS-AS1_ directly binds to PACT**. To explore the underlying mechanism through which *DARS-AS1* regulates cell proliferation, we performed an RNA pull-down assay to identify its potential protein binding partners. RT-qPCR results show that about 86.2% of *DARS-AS1* locates in the cytoplasm of SW620 cells (Supplementary Fig. 3a). Then, in vitro transcribed biotinylated *DARS-AS1* or mock RNA were incubated with SW620 cell lysates, followed by SDS-PAGE separation. Subsequent silver staining shows a distinct band (~38 kDa) is significantly enriched in the *DARS-AS1* pull-down sample but not mock RNA or beads samples (Fig. 3a). This band was identified as PKR-activating protein (PACT) by mass spectrometry (MS), and further validated by immunoblotting in SW620, HCT116 and HepG2 cell lines (Fig. 3a, b). The enrichments of DARS and PACT associated proteins-PKR and TRBP-were also examined by RNA pull-down via western blot (WB). As results show, no direct interactions between *DARS-AS1* RNA and these three proteins were detected (Supplementary Fig. 3b). The specific interaction between *DARS-AS1* and PACT was further confirmed by RNA immunoprecipitation (RIP) assay, which show that *DARS-AS1* is significantly enriched by anti-PACT antibodies, while not other control RNAs (Fig. 3c). To determine whether *DARS-AS1* directly interacts with PACT in the absence of any other cellular component, in vitro biolayer interferometry (BLI) assays were performed using purified PACT. Biotin-labeled *DARS-AS1* or mock RNA were immobilized on streptavidin (SA) biosensors, following by incubating into kinetic buffer containing 1 µM PACT. Notably, PACT strongly bound to *DARS-AS1* ($K_D$ values about 26.9 nM) but not to the mock RNA (Fig. 3d). Together, these results demonstrate a direct interaction and high affinity between *DARS-AS1* and PACT.

We then generated three biotinylated RNA fragments of *DARS-AS1* by in vitro transcription to identify the region in *DARS-AS1* required for PACT association (Fig. 3e). The RNA pull-down results suggest that each fragment is able to interact with PACT, but that the 3' end region (384–768nt, marked A3) shows a stronger association than the middle region (192–576nt, marked A2) or the 5' end region (1–384nt, marked A1) (Fig. 3e). Similar results were observed in the in vitro RIP assay using recombinant PACT (Fig. 3f). Consistent with these results, binding experiments of immobilized RNA fragments to PACT using BLI also show that PACT has a higher affinity with A3 (384–768nt) ($K_D$ values about 94.6 nM), while barely binds to other regions. (Fig. 3g).

We also investigated the associated binding region in PACT. PACT comprises three functional domains, with two conserved double-stranded RNA-binding domains (dsRBD) and a third domain (marked D3), which serves as an activator for interacting proteins. To investigate the lncRNA binding capability of each domain, we constructed three mutations with a respective deletion of each of these three domains and performed in vitro RIP analysis. Our results show that deletion of the third domain (D3) of PACT dramatically reduces its interaction with *DARS-AS1* (0.11-fold of intact PACT) (Fig. 3h) as compared to the other two mutations, highlighting D3 as the crucial domain for interaction with *DARS-AS1*. Taken together, these results suggest that the interaction between *DARS-AS1* and PACT might occur primarily through the 3' end of *DARS-AS1* and D3 domain of PACT.

**_DARS-AS1_ promotes cancer cell proliferation and inhibits cell apoptosis through inhibiting the function of PACT**. We noticed that, *DARS-AS1* has no influence on the expression of PACT, neither does PACT affect *DARS-AS1* (Supplementary Fig. 3c). We next detected the effect of knocking down PACT on cell growth. On the contrary with *DARS-AS1*, the relative cell growth was 1.5 to 3 times faster when knocking down PACT (Supplementary Fig. 3d). The results of colony formation assay showed cells formed 2 to 3-fold more colonies after PACT shRNA treatment (Supplementary Fig. 3e). To validate whether *DARS-AS1* regulates cell proliferation through PACT, we generated SW620 cells overexpressing PACT, *DARS-AS1*, or both. Overexpression of PACT shows notably repression of cell proliferation (Fig. 3i). While solely overexpressing *DARS-AS1* significantly promotes cell proliferation, the growth rate of cells overexpressing both *DARS-AS1* and PACT shows no significant difference. These results indicate that PACT could counteract the enhanced proliferation induced by the overexpression of *DARS-AS1*.

As different regions of *DARS-AS1* have variant binding capabilities with PACT, we examined their relative impacts on cell proliferation by variably overexpressing *DARS-AS1* fragments. Overexpression of the 3' end of *DARS-AS1* (384-768nt), which is the major region in *DARS-AS1* associating with PACT, exhibits the strongest ability to promote cell proliferation compared with the other two fragments (Fig. 3j). These results indicate a positive association between binding capability and the biological function of *DARS-AS1*.

It has been reported that PACT is a proapoptotic protein[19]. We therefore examined the effect of *DARS-AS1* on apoptosis. As expected, knockdown of *DARS-AS1* dramatically enhances the cleavage of PARP in SW620 cells and increases the proportion of AnnexinV-positive cells in SW620, HCT116, HepG2 and MBA-MD-231 cell lines (Fig. 3k, l and Supplementary Fig. 3f–h), indicating the anti-apoptotic role of *DARS-AS1* in cancer cells,

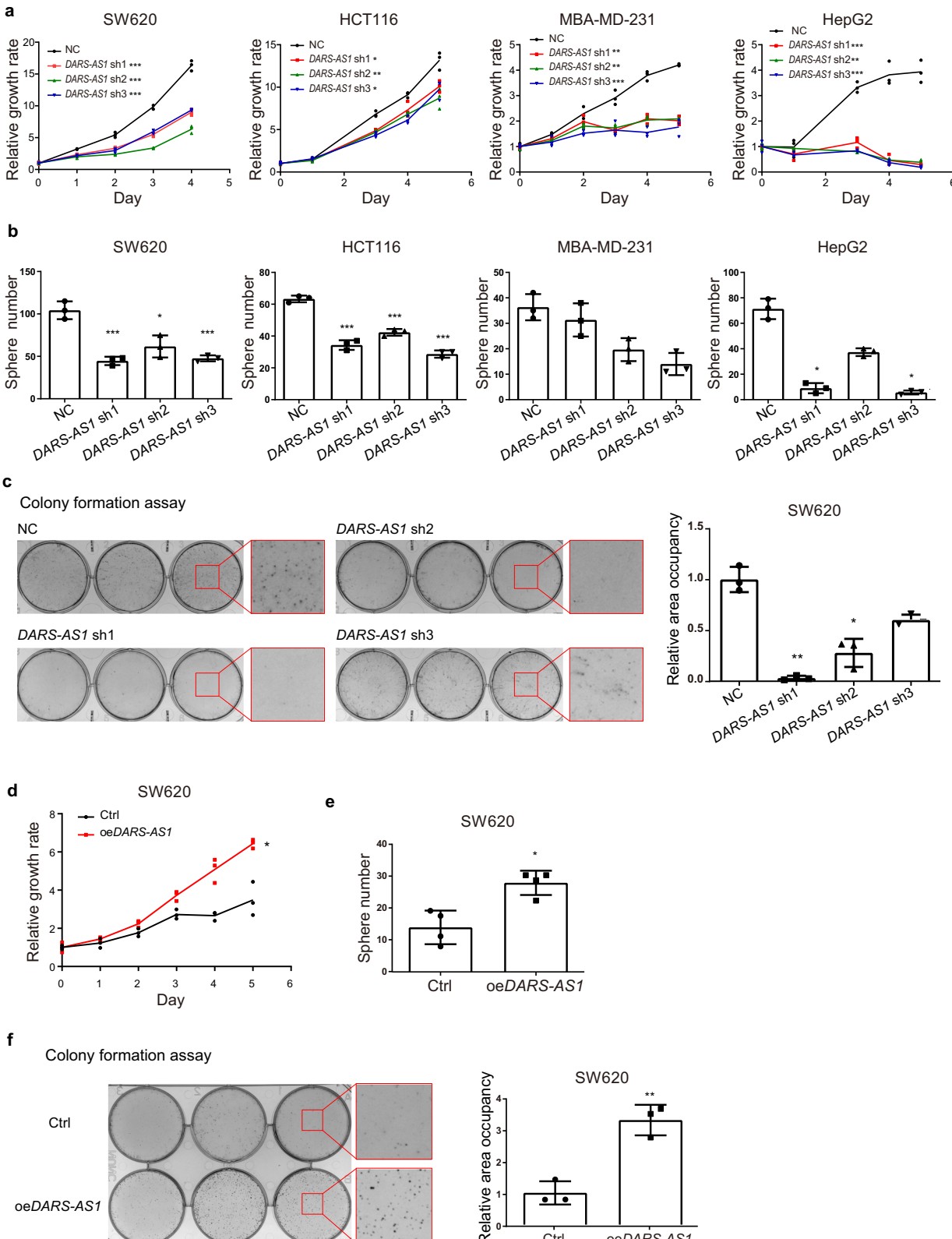

**Fig. 2 Knockdown of *DARS-AS1* suppresses cell proliferation in multiple cancers.** Impacts on cell proliferation (**a**) and sphere formation (**b**) of SW620, HCT116, MBA-MD-231, and HepG2 cells by control shRNA and *DARS-AS1*-shRNAs. **c** Impacts on colony formation of SW620 cells by control shRNA and *DARS-AS1*-shRNAs. Cell proliferation (**d**), sphere formation (**e**) and colony formation (**f**) of *DARS-AS1*-overexpressing SW620 cells. Data shown are means ± SD in triplicates experiments. *$p \leq 0.05$, **$p \leq 0.01$ and ***$p \leq 0.001$, by two-tailed Student's *t* test.

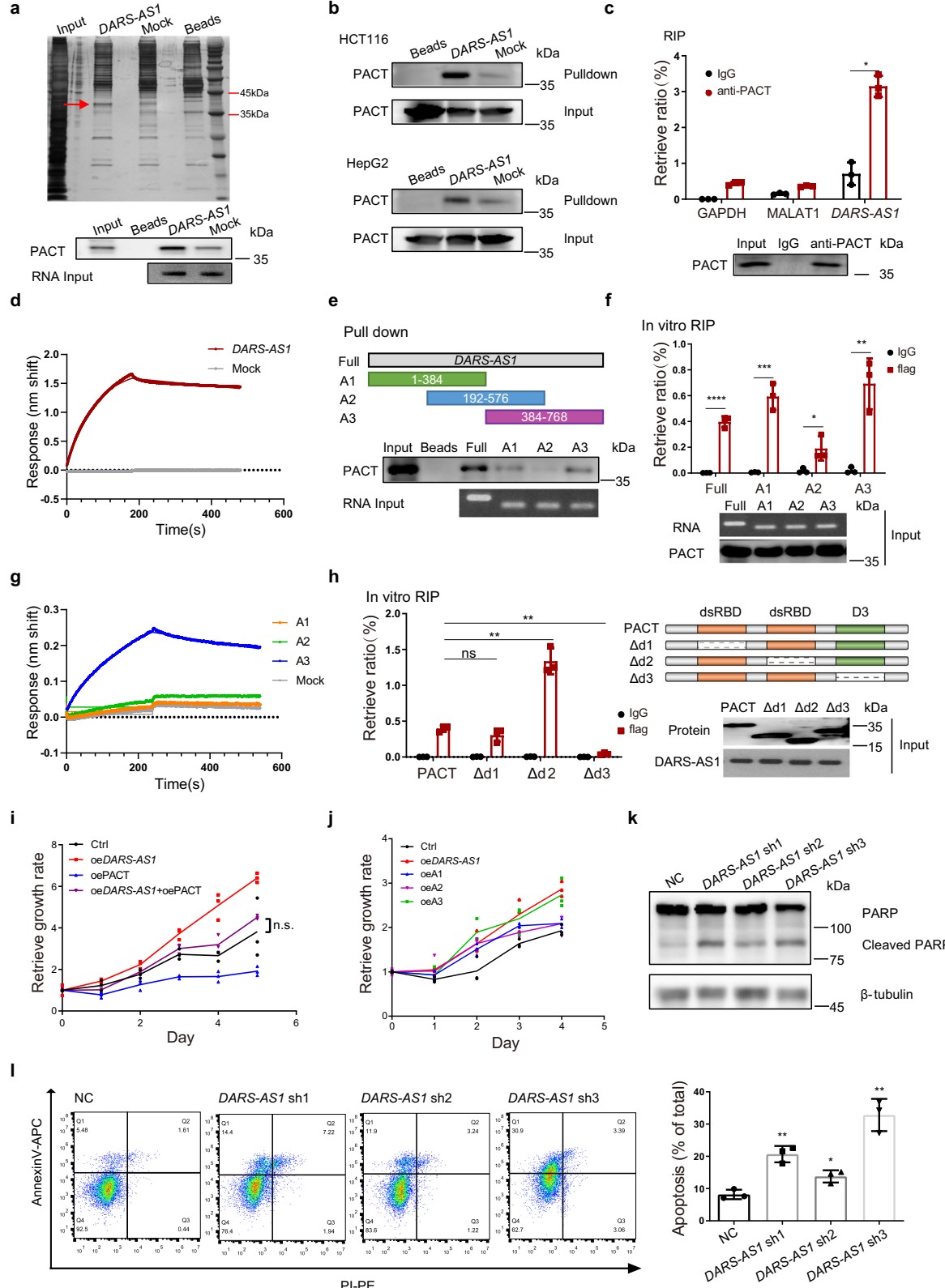

contrary to the apoptosis-inducing function of PACT. Together, these results suggest that the mechanism of oncogenic function for *DARS-AS1* might be through inhibiting the function of PACT.

**DARS-AS1 blocks the interaction between PACT and PKR.** We further explored the functional consequences of *DARS-AS1*-PACT association. It has been reported that PACT activates PKR

by direct interaction, which subsequently enhances eIF2α phosphorylation, inducing translation abolishment and apoptosis[17]. Firstly, we detected if *DARS-AS1* influences the cellular location of PACT and PKR. As observed by fluorescence confocal microscope, PACT and PKR are highly colocalized in SW620 cells with a mean Pearson's correlation coefficient of 0.72. While, overexpression of *DARS-AS1* significantly reduces

**Fig. 3 *DARS-AS1* directly interacts with PACT. a** RNA pull-down assay identifies *DARS-AS1* interacting with PACT in SW620 cells. Upper, silver staining of associated proteins. Lower, immunoblot with anti-PACT antibodies. **b** RNA pull-down assays were performed in HCT116 (upper) and HepG2 (lower) cells. Immunoblot detected the enrichment of PACT. **c** RNA immunoprecipitation (RIP) assays were performed using the indicated antibodies in SW620 cells. **d** The binding curves of PACT with full length *DARS-AS1* or control RNA were obtained by biolayer interferometry (BLI). RNAs were immobilized on streptavidin biosensors. 1 μM PACT were used for measuring the association. **e** RNA pull-down assays were performed using biotinylated full length *DARS-AS1* or truncations (upper). Immunoblot shows the retrieved PACT (lower). **f** Purified flag-tagged PACT was incubated with biotinylated full length *DARS-AS1* or truncations (as showed in **e**) for in vitro RIP assays. Retrieved RNAs were validated by RT-qPCR. **g** The relative affinities of different RNA fragments for PACT were obtained by biolayer interferometry. 100 nM RNA and 1 μM PACT were used in each assay. **h** In vitro RIP assays were performed using purified intact or truncated flag-tagged PACT. Retrieved RNA was validated by RT-qPCR. **i** Cell growth rate of SW620 cells overexpressing *DARS-AS1*, PACT, or both. **j** Overexpression of full length or truncated *DARS-AS1* in SW620 cells shows different impacts on cell growth. **k** Cell apoptosis was detected by immunoblot with anti-PARP antibodies. **l** Knockdown *DARS-AS1* induces SW620 cell apoptosis, as revealed by flow cytometry. Data shown are means ± SD in triplicates experiments. *$p \leq 0.05$, **$p \leq 0.01$, ***$p \leq 0.001$, ****$p < 0.0001$, by two-tailed Student's *t* test.

the colocalization of PACT and PKR (mean Pearson's correlation coefficient of 0.61) (Fig. 4a). To investigate whether *DARS-AS1* can regulate the interaction of PACT-PKR, we performed co-immunoprecipitation (co-IP) analysis with anti-PACT antibody in SW620 cell lysates. PKR is strongly enriched by anti-PACT in control cells, while the retrieval rate of PKR in lysate from *DARS-AS1*-overexpressing cells obviously decreases (Fig. 4b). Purified flag-tagged PACT and PKR were used for in vitro protein binding analysis. Consistently, those supplied with *DARS-AS1*, but not control RNA, exhibits an inhibited PACT-PKR interaction (Fig. 4c). All results indicate that *DARS-AS1* interrupts the association of PACT and PKR.

***DARS-AS1* represses the PACT-mediated phosphorylation of PKR**. It is generally believed that PKR phosphorylation at Thr451 could be induced, once PACT interacts with PKR[17]. Our results demonstrate that the phosphorylation level of PKR is significantly increased in *DARS-AS1*-knockdown cells after serum starvation (Fig. 4d and Supplementary Fig. 4a). Congruently, we detected that the phosphorylation of eIF2α—a main substrate of PKR—is also dramatically increased by *DARS-AS1* shRNAs (Fig. 4d and Supplementary Fig. 4a). Thapsigargin is an ER stressor, which causes Ca$^{2+}$ release from ER. It is reported that thapsigargin treatment induces PACT expression and activation, which further interacts with and activates PKR, resulting in cell apoptosis via increasing eIF2α phosphorylation[18,61]. Here, we used thapsigargin as a stimulator of PACT/PKR pathway to investigate whether *DARS-AS1* can help cells overcome stress through suppressing PACT/PKR pathway. We observed a positive correlation between the expression level of *DARS-AS1* and cell resistance to thapsigargin. SW620 cells overexpressed *DARS-AS1* survived better under thapsigargin treatment, while *DARS-AS1*-knockdown cells became more sensitive (Fig. 4e). In agreement with these results, overexpression of *DARS-AS1* recedes the phosphorylation of PKR raised by thapsigargin (Supplementary Fig. 4b). In contrast, after thapsigargin treatment, PKR and eIF2α in *DARS-AS1*-knockdown cells were more phosphorylated compared with control cells (Supplementary Fig. 4b). Interestingly, thapsigargin induces *DARS-AS1* expression in dose-depended manner, which may imply the anti-stress function of *DARS-AS1* (Supplementary Fig. 4c). Additionally, we performed in vitro activation assays to validate these observations. Briefly, PKR was purified from cell lysates using anti-PKR antibodies, then incubated with recombinant PACT and in vitro transcribed *DARS-AS1*. After enzymatic reaction, phospho-PKR was detected by WB. Our results exhibit that the phosphorylation of PKR is clearly inhibited by *DARS-AS1*, but not control RNA (Fig. 4f). These in vitro and in vivo results suggest that *DARS-AS1* represses PACT-mediated PKR activation. Meanwhile, we

also observed that the retrieval of PACT is reduced in the presence of *DARS-AS1* (Fig. 4f). This result is consistent with results of in vitro protein binding analysis (Fig. 4c), and again illustrates a blocker-like function of *DARS-AS1* on PACT-PKR association.

Ser246 and Ser287 in the D3 domain of PACT is essential to activate PKR under cell stresses[22]. Substitution of both serine residues by alanine produces a mutant PACT (mutD) which activates PKR without stresses, while replaced by alanine (mutA) abolished the activation ability of PACT. Since we have demonstrated this domain's importance in the direct association with *DARS-AS1*, we generated these two PACT mutants to examine whether these residues may also be involved in the interaction with *DARS-AS1*. Intriguingly, both mutants lose binding capacity to *DARS-AS1* (Supplementary Fig. 4d), indicating that the intact PACT protein structure might be required for effective interaction with *DARS-AS1*.

Moreover, our results further show that the inhibition of cell proliferation caused by *DARS-AS1*-shRNAs could be partially recovered through the overexpression of either dominant negative PACT mutant (PACTmutA) or dominant negative PKR mutant (PKRmut) (Supplementary Fig. 4e, f). The overexpression of dominant negative PKR mutants decreases the *DARS-AS1*-knockdown induced PKR phosphorylation in serum starved cells as well as the phosphorylation of eIF2α (Fig. 4g). What's more, the proportion of apoptotic cells caused by *DARS-AS1*-knockdown was also reduced in PKRmut-overexpression cells (Fig. 4h and Supplementary Fig. 4g). Inhibiting kinase activity of PKR also weakens the function of *DARS-AS1*, as results show that knocking down *DARS-AS1* rarely trigger the phosphorylation of PKR and eIF2α when treating cells with C16, a PKR specific inhibitor (Fig. 4i and Supplementary Fig. 4h). Taken together, our results suggest that the promotion of cell proliferation by *DARS-AS1* is, at least partially, through repressing PACT-mediated PKR activation.

**High *DARS-AS1* expression promotes tumor progression in vivo and correlates with poor clinical outcomes**. To further examine the role of *DARS-AS1* in tumorigenesis, we performed in vivo experiments using xenograft mouse models. Results show that knockdown of *DARS-AS1* dramatically decreased tumor growth in mice ($p$ value < 0.0001) (Fig. 5a). Accordingly, significant reductions of about 72.9% of mean tumor volumes and about 87.8% of mean tumor weights in the *DARS-AS1*-knockdown group were observed (Fig. 5b–d). These results strongly suggest that *DARS-AS1* could significantly promote tumor growth in vivo.

To gain further insight into the clinical impact of *DARS-AS1*, we investigated the correlation between its expression and patients' survival. By analyzing TCGA datasets, we found that a higher expression of *DARS-AS1* is significantly correlated with

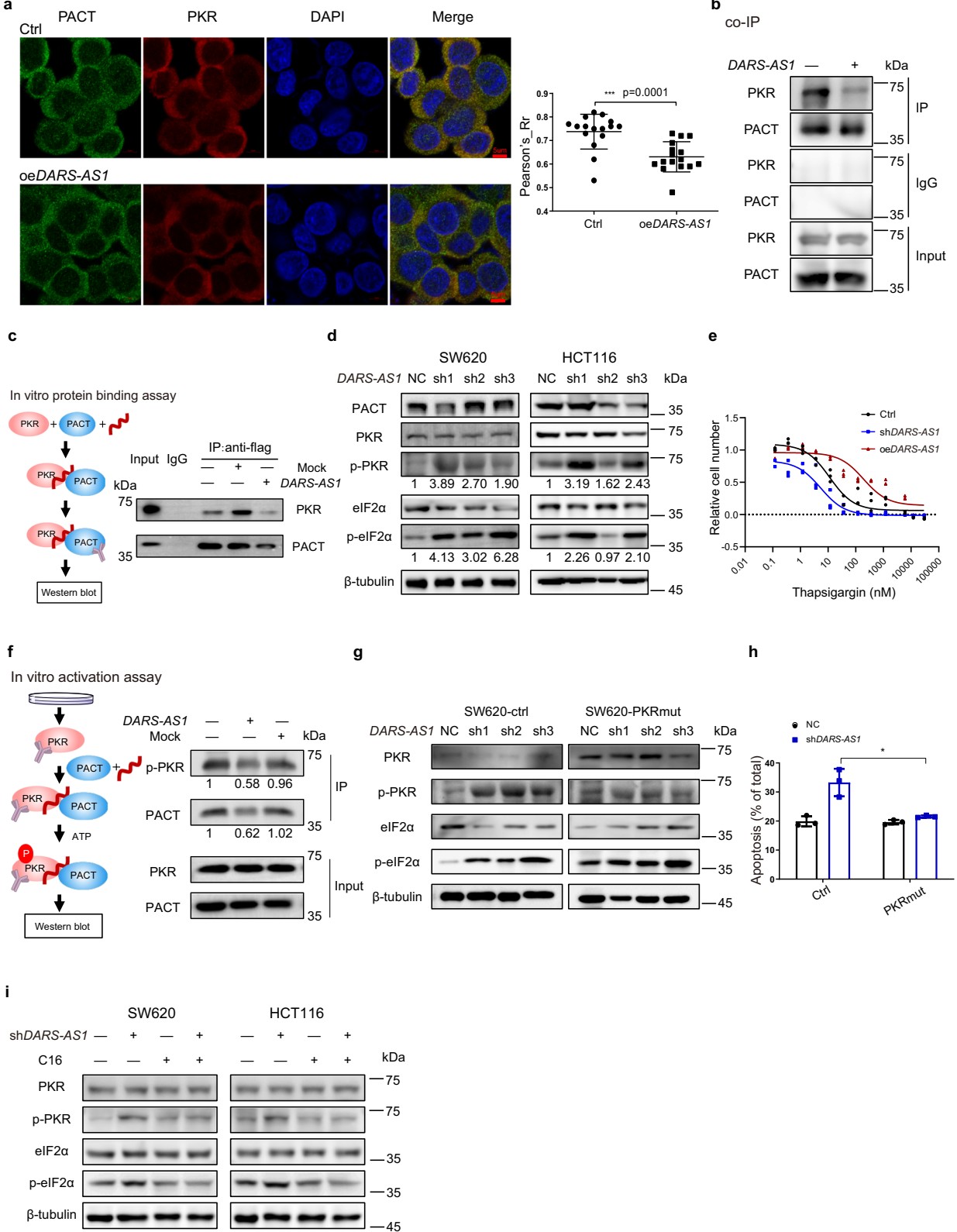

poorer survival in uveal melanoma (UVM), kidney chromophobe (KICH), kidney renal papillary cell carcinoma (KIRP), mesothelioma (MESO), glioblastoma multiforme (GBM), and brain lower grade glioma (LGG) patients (Fig. 5e). These results indicate that *DARS-AS1* might play an important role in clinical tumor progression and might be a potential prognosis biomarker for multiple cancers.

## Discussion

In this study, using large-scale CRISPRi functional screening, we identified lncRNA *DARS-AS1* as a player in overcoming cancer cellular stresses through modulating two key stress responders, PACT and PKR. Through direct interaction with PACT, *DARS-AS1* inhibits PACT-mediated PKR activation, thus prevents apoptotic cell death and promotes cell proliferation (Fig. 5f). The

**Fig. 4 *DARS-AS1* blocks PACT-mediated PKR activation. a** The colocalization of PACT and PKR in control cells or *DARS-AS1*-overxpression cells were observed by fluorescence confocal microscope. Cell nucleus was stained by DAPI. The statistic results were obtained by 16 photos. **b** Co-immunoprecipitation (co-IP) was performed using anti-PACT antibody in cell lysates of SW620 control cells or *DARS-AS1*-overexpressing cells. **c** Flag-tagged PACT, purified PKR, and in vitro transcribed *DARS-AS1* or mock RNA were incubated for in vitro protein binding assay. Immunoprecipitation used anti-flag antibody. **d** Immunoblot using indicated antibodies was performed in SW620 and HCT116 cells transfected with control shRNAs or *DARS-AS1*-shRNAs, followed by serum starvation. **e** The expression level of *DARS-AS1* alters cell sensitivity of thapsigargin. SW620 cells were transfected with *DARS-AS1* shRNAs, *DARS-AS1*-overexpression plasmids or control plasmids. Cells were treated by thapsigargin for 48 h and detected cell viability by MTS reagents. **f** In vitro transcribed *DARS-AS1* or mock RNA and purified PACT were used for in vitro activation assay and immunoblot detection. **g** Immunoblot using indicated antibodies were performed in SW620-ctrl cells (left) or cells overexpressing PKR mutant (right). These cells were then transfected with control shRNAs or *DARS-AS1*-shRNAs, followed by serum starvation. **h** Inactive mutant PKR compensated SW620 cell apoptosis induced by *DARS-AS1*, as revealed by flow cytometry. **i** Immunoblot using indicated antibodies were performed in SW620 (left) or HCT116 cells (right). Cells transfected with control shRNAs or *DARS-AS1*-shRNAs were treated by serum starvation, suppling with 100 nM PKR inhibitor C16 or DMSO. Scale bars =5 μm. Data shown are means ± SD in triplicates experiments. *$p \leq 0.05$, by two-tailed Student's $t$ test.

upregulation of *DARS-AS1* is observed in numerous cancer types, suggesting its function to promote cancer cell survival under stress conditions may broadly apply to a multitude of carcinomas.

PACT has been identified as a protein activator of PKR, and PACT-mediated PKR activation plays an important role in stress responses through regulating transcription, translation, apoptosis, and other essential cellular processes[62]. For decades, efforts have been made to understand the cancer-specific regulation of PACT-PKR cascade[63]. Here, our study discloses another mechanism underlying PACT-PKR regulation in cancer cells through the cellular lncRNA *DARS-AS1*, which directly binds to PACT, blocks PACT-PKR interaction, suppresses PKR activation and eIF2α phosphorylation, consequently inhibits stress-induced apoptosis and promotes cancer cell proliferation eventually. This discovery elucidates a potential lncRNA target for cancer prognosis and therapy.

Our data shows that the knockdown of *DARS-AS1* sensitizes cells to serum starvation, with a significant increase in phosphorylated PKR and eIF2α. These results suggest that *DARS-AS1* facilitates cancer cells survival in severe environments through inhibiting PACT/PKR activity. Some other non-coding RNAs, such as *ASPACT* and *nc886*, have also been found to be involved in the PACT/PKR axis, by reducing the mRNA level of PACT[48] or regulating autophosphorylation through binding to PKR[49,50,64]. Among these, *DARS-AS1* acts as an interferer of PACT-PKR association. This study enriches our understanding of the regulation of the PACT/PKR axis and the role of lncRNAs in stress responses.

PACT contains three independent domains. Each of the first two dsRBDs is sufficient for high-affinity binding of PACT to PKR, and the third domain (D3) is required for PKR activation in vitro and in vivo[65]. Our research found that *DARS-AS1* preferentially interacts with the D3 domain (Fig. 3h). Considering the large size of lncRNAs (768nt), the binding of *DARS-AS1* to D3 might physically inhibit the interaction between the PACT dsRBD domains and PKR, thus blocking the association of PACT and PKR. Point mutations of PACT with substituted Ser246 and Ser287 in D3 by alanine or aspartic acid destroy its binding affinity to *DARS-AS1*, indicating the importance of the intact structural and electrical properties of D3 in their association. Future work using more precise biochemical assays and high-resolution structural analysis of PACT will be needed to further understand the details of this mechanism.

Prior studies have reported that *DARS-AS1* promotes cell proliferation by variety mechanisms[51–53]. In one example, researchers observed that *DARS-AS1* upregulated the expression of its antisense protein-coding gene *DARS*, by targeting *miR-194-5p* in renal cancer cells[53]. However, in the present study, the knockdown of *DARS-AS1* shows little effect on the transcription of *DARS* across multiple cancer types at least including colorectal cancer, breast cancer and liver cancer. As lncRNAs display cell and tissue specific expression patterns, functional mechanisms may not be conserved across different carcinoma types, which might cause this discrepancy between our observations and previous evaluations on different cancers. Specific studies will be needed to elucidate the particular mechanisms in different physiologic and pathological processes.

Analysis of clinical data revealed a negative correlation between *DARS-AS1* expression in tumors and the survival of cancer patients, underscoring the significance of *DARS-AS1*/PACT/PKR axis in cancer prognosis. In summary, our study illustrates that *DARS-AS1* is a regulator of PACT/PKR signaling axis, which promotes cancer cell proliferation and inhibits cell apoptosis during stress responses, providing another line of inquest and an exciting potential therapeutic for future study.

## Methods

**Cell lines**. Human cell lines including SW620, A549, MBA-MD-231, HCT116, HepG2 and HEK293T were obtained from National Infrastructure of Cell Line Resource of China. All cells were maintained in DMEM medium (DMEM, Thermo Fisher Scientific, Waltham, MA) supplied with 10% FBS (Gemini, Brooklyn, NY) and 1% Penicillin-Streptomycin Solution (Thermo Fisher Scientific), at 37 °C in a 5% $CO_2$ incubator.

**Antibodies**. The antibodies used in this paper were listed: anti-PACT, Abcam (ab31967); anti-PKR, Abcam (ab184257); anti-PKR (phospho T451), Abcam (ab81303); anti-flag, Abcam (ab125243); anti-eIF2α, Abclonal (A0764); anti-eIF2α (phospho S51), Abcam (ab32157); anti-PACT (phosphor S246), Abgent (AP7744b); anti-β-tubulin, CST (2128); Normal mouse IgG, CST (5415S); Normal rabbit IgG, CST (2729S). Antibodies were diluted as 1:1000 with PBST for western blotting, and 1:100 for IP.

**Pooled sgRNA library construction, virus packaging, and screening**. The sgRNAs were designed using a public tool named CRISPR-ERA[66]. We utilized the default parameters of this tool to design the sgRNAs and the algorithm computes sgRNA binding sites within a 3 kb region centered at TSS. The pool of sgRNA oligos was synthesized at CustomArray, Inc. (Bothewell, WA) and cloned into a pgRNA-humanized plasmid (Addgene #44248). In total, 12 μg pooled pgRNA-humanized plasmids, 7.2 μg psPAX2 (Addgene #12260), and 4.8 μg pMD2.G (Addgene # 12259) were co-transfected into 5 × 10⁶ HEK293T cells in a 10 cm dish using DNAfect transfection reagent (CWBIO, Beijing, China), by following the manufacturer's instructions. Supernatant containing virus particles were collected at 48 h and 72 h after transfection and filtered through 0.45 μm filters. For screening, SW620 cells expressing dCas9/KRAB fusion proteins were generated by viral transduction. Modified SW620 cells were infected with the viral library in four independent infection experiments at MOI 0.1–0.3 and selected with 2 μg/ml puromycin (Sigma, St. Louis, MO) over 2 days. After, cells were in vitro cultured for 18 days at minimum library coverage of 500 cells/sgRNA for screening.

**sgRNA library construction for next generation sequencing**. Genomic DNA was extracted following the instructions of QIAamp DNA Blood Midi Kit (QIA-GEN, Dusseldorf, Germany; 51183). In total, 100 μg genomic DNA of each biological replicate was used for library construction. sgRNA region was amplified and incorporated with a barcode by two-round PCR.

First-round PCR primers:

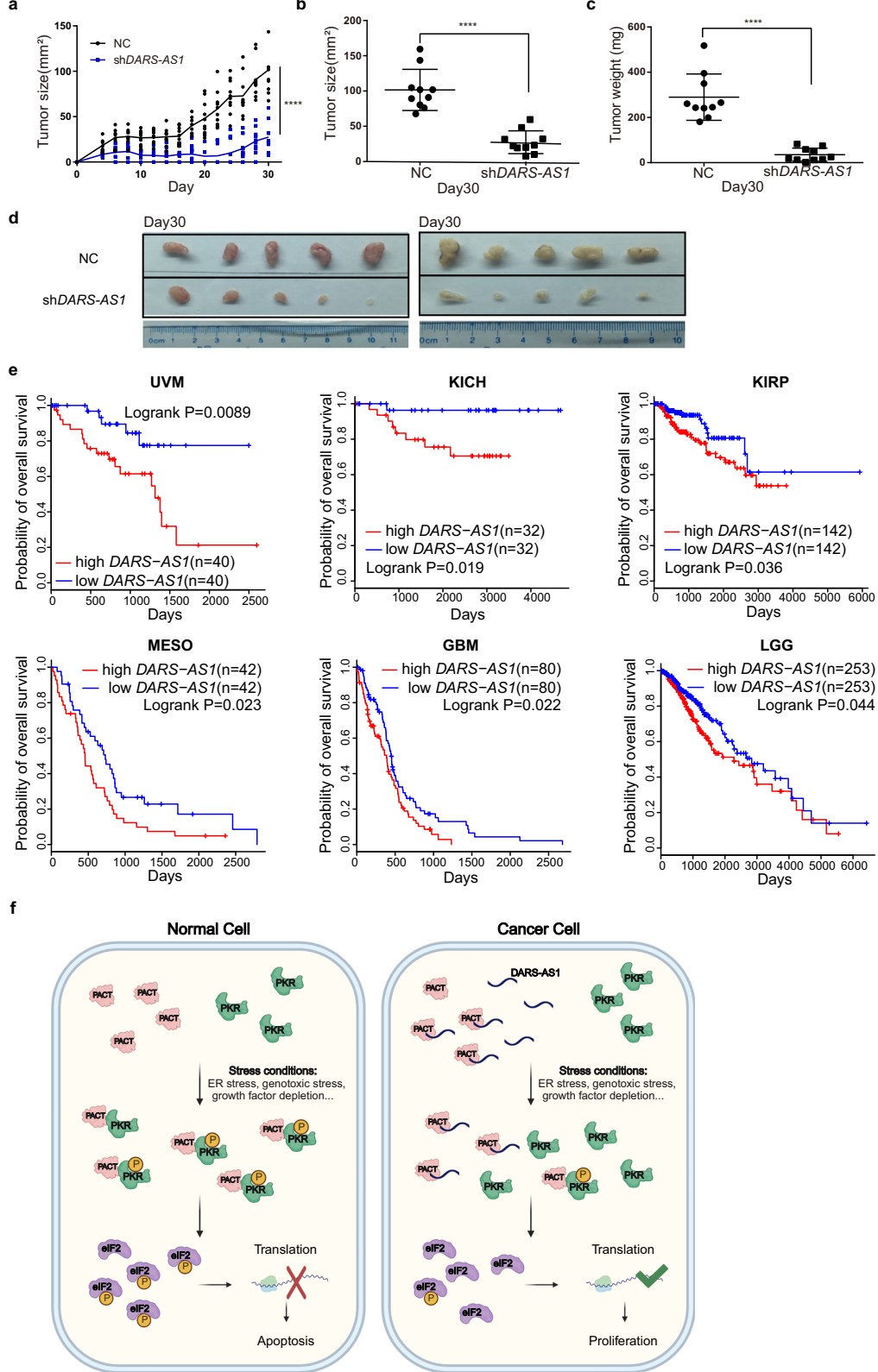

**Fig. 5 *DARS-AS1* promotes tumor proliferation in vivo and correlates with poor prognosis. a–d** Effect of *DARS-AS1*-knockdown on colorectal tumorigenesis in nude mice. Growth curve (**a**), tumor size (**b**), weight (**c**) and tumor image (**d**) are shown. Error bars represent ±SEM. $n = 10$. ****$p < 0.0001$, by two-tailed Student's $t$ test. **e** Kaplan–Meier analyses of the correlations between *DARS-AS1* expression level and overall survival of patients with UVM, KICH, KIRP, MESO, GBM, and LGG. High, patients in the top 50% of *DARS-AS1* expression levels; low, patients in the bottom 50% of *DARS-AS1* expression levels. $p$ values were determined using a log rank test. **f** Proposed model of *DARS-AS1* in regulating PACT-PKR pathway and tumor growth.

F,ACACGACGCTCTTCCGATCTNNNNN(randomsequence)TGATAACGG
ACTAGCCTTATTTTAACTTGC;

R,GAGTTCAGACGTGTGCTCTTCCGATCTGTTGATAACGGACTAGCCT
TATT.

Second-round PCR primers:

F,AATGATACGGCGACCACCGAGATCTACACACACTCTTTCCCTACAC
GACGCTCTTCCGATCT;

R,CAAGCAGAAGACGGCATACGAGATNNNNNNN(barcode)
GTGACTGGAGTTCAGACGTGTGCTCTTCCGATCT.

PCR products were purified using NucleoSpin® Gel and PCR Clean-up kit
(MACHEREY-NAGEL, Düren, Germany; 740609.250) and quantified with Qubit™
dsDNA HS Assay Kit (Thermo Fisher Scientific; Q32854).

**MTS assay**. MTS assays were used to measure cell proliferation. Cells were seeded
in a 96-well-plate at an initial density of 2000 cells/well. Relative cell number was
detected at a specified time each day for total 4–6 days. For each well, 20 μl MTS
reagent (Promega) was diluted with 100 μl DMEM, incubated with cells at 37 °C for
4 h, and then measured by OD490.

**Sphere formation assay**. The unanchored growth ability was detected by sphere
formation assay. Briefly, 2000 cells transfected with *DARS-AS1* shRNAs or control
shRNAs were cultured on ultra-low attachment microplates (Corning), with the
media changed every 4 days. Spheres were counted after 14 days. For the gain-of-
function assay, 500 cells transfected *DARS-AS1*-overexpressing plasmid or control
plasmid were used instead and otherwise the methodology was unchanged.

**In vitro RNA transcription**. RNA was transcribed using T7 RNA polymerase and
biotin-16-UTP (Roche 1138908910) following the instructions of Riboprobe®
Combination Systems (Promega P1440). Primers used here are listed in Supple-
mentary Table 4.

**Protein expression and purification**. The protein-coding region of PACT or PKR
was cloned into pET15b (Addgene #73619) and transformed in BL21(DE3). The
bacteria were incubated in LB supplied with ampicillin overnight, then diluted 100-
fold in fresh LB. Protein expression was induced by adding of 1 mM IPTG when
the OD600 of the culture medium reached 0.8. After overnight incubation under
gentle shaking (250 rpm at 20 °C), cell pellets were harvested by centrifugation
(4000 rpm, 10 min, 4 °C). Cell pellets were resuspended in lysis buffer (50 mM Tris
pH 8.0, 250 mM NaCl, 1 mM PMSF) and incubated on ice for 30 min, followed by
sonication (15 min, 5 s on/off, on ice) and centrifugation (13,000 rpm, 30 min,
4 °C). The supernatant was then applied to Ni-NTA resin (QIAGEN) 3 times at
4 °C, washed 4 times with a washing buffer (50 mM Tris pH 8.0, 40 mM imidazole,
250 mM NaCl), and eluted 3 times with a total of 10 ml elution buffer (50 mM Tris
pH 8.0, 250 mM NaCl, 300 mM imidazole). The purified protein was detected by
WB and the concentration was determined by Qubit™ Protein Assay Kit (Thermo
Fisher Scientific; Q 33212).

**RNA immunoprecipitation (RIP)**. RIP assay was performed as previously
described[67] with modifications. Briefly, $1 \times 10^7$ cells were lysed with 1x RIP buffer
(25 mM Tris-HCl, pH 7.5, 100 mM NaCl, 0.5% NP-40, RNasin ribonuclease
inhibitor (Promega), PMSF (Beyotime Biotechnology), 1 mM DDM, protease
inhibitor cocktail (Roche), 1 mM DTT) and centrifuged at 13,000 rpm for 15 min at
4 °C. Then, the supernatant was incubated with protein A + G magnetic beads
(Millipore) conjugated 5 μg anti-PACT antibodies (Abcam) or IgG (CST). The
beads were washed 5 times with 5 × RIP buffer, and then digested by proteinase K
(NEB). RNA was extracted with Trizol and detected by RT-qPCR. Primers are
provided in Supplementary Table 5.

**In vitro RIP**. In vitro RIP assay was performed following a modified protocol of a
standard RIP assay. In total, 5 pmol in vitro transcribed RNAs were diluted with
1×RIP buffer and renatured by incubating for 5 min at 65 °C, then slowly cooling
down to room temperature. In total, 5 pmol intact or mutant flag-tagged PACT
proteins purified from *E. coli.* were incubated with renatured RNA for 2 h at 4 °C,
and used for IP with anti-flag antibodies following the procedure of RIP assay
described above.

**RNA pull-down**. For RNA pull-down assay, $1 \times 10^7$ cells were lysed with 1× RIP
buffer. Following centrifuging at 13,000 rpm for 15 min at 4 °C, the supernatant
were pre-treated with 30 μl streptavidin magnetic beads (Beckman) for 2 h at 4 °C.
The cleaned lysates were then supplied with yeast tRNA and incubated with
40 pmol renatured RNA overnight at 4 °C, followed by incubating for additional
2 h supplied with 20 μl new BSA-blocked streptavidin magnetic beads. The washing
step contains 4 times with 5×RIP buffer and 4 other times with 1× RIP buffer.
Associated proteins were eluted with biotin elution buffer (25 mM Tris-HCl, pH
7.5, 12.5 mM D-biotin, PMSF) and resolved on NuPAGE 4-12% Bis-Tris Gel
(Invitrogen). After silver staining (Beyotime Biotechnology), specific bands were
cut and analyzed by MS.

**Co-immunoprecipitation (co-IP)**. Co-IP assay was performed to test the inter-
action between PACT and PKR. Briefly, supernatant lysates were prepared by
incubating $1 \times 10^7$ lysed cells in 1 × RIP buffer, then centrifuging at 13,000 rpm for
15 min at 4 °C. Lysates were supplied with protein A + G magnetic beads con-
jugated 5 μg anti-PACT antibodies and gently rotated overnight at 4 °C. Beads were
washed three times with 5 × RIP buffer and twice with 1 × RIP buffer, followed by
elution with 1 × SDS buffer. The retrieved proteins were analyzed via SDS-PAGE
gel and detected by WB.

**In vitro protein binding assay**. Two pmol flag-tagged PACT and 1 pmol PKR
purified from *E. coli.* were diluted in 1 × RIP buffer and incubated with 10 pmol
renatured RNA for 2 h at 4 °C. After, they were incubated two additional hours
with protein A + G magnetic beads conjugated anti-flag antibodies. Beads were
then washed four times with 1 × RIP buffer and eluted by 1 × SDS buffer. Retrieved
PACT and PKR were detected by WB.

**In vitro activation assay of PKR**. In total, $1 \times 10^7$ Hela cells were lysed with
1 × RIP buffer and centrifuged at 13,000 rpm for 15 min at 4 °C. PKR was purified
from the supernatants using protein A + G magnetic beads conjugated anti-PKR
antibody: washed four times with 5 × RIP buffer, once with 1 × RIP buffer, and then
resuspended in 1 × RIP buffer. Purified PACT and in vitro transcribed RNA were
incubated with beads for 2 h at 4 °C, then washed three times with 1 × RIP buffer,
once with kinase buffer (20 mM HEPES at pH 7.4, 40 mM MgCl$_2$, 100 mM KCl,
1 mM DTT, PMSF, 0.5% NP-40, protease inhibitor cocktail, RNasin ribonuclease
inhibitor, phosphatase inhibitor cocktail (Beyotime Biotechnology)), then resus-
pended in kinase buffer containing 50 μM ATP (NEB). After 30 min at 30 °C the
reaction was terminated through a wash with 1 × RIP buffer. Proteins were eluted
by 1 × SDS buffer and resolved on SDS-PAGE gel. Phospho-PKR were
detected by WB.

**Immunofluorescence**. Cells in confocal dishes were washed with PBS and fixed
with 4% formaldehyde for 10 min. Permeabilized in 0.1% Triton X-100 for 10 min
and then blocked with 1% BSA in 0.1% PBS-Tween 20 for 1 h. Cells were incubated
with anti-PACT antibody (Abcam ab31967, 5 μg/ml) (1% BSA, 0.1% Tween 20,
PBS) for 2 h at room temperature followed by washing with PBS for three times.
The secondary antibody labeled with Alexa Fluor® 488 (Abcam ab150077) was
used at a 1/1000 dilution for 1 h. After washing with PBS for three times, cells were
incubated overnight at 4 °C with anti-PKR (Alexa Fluor® 647) antibody (Abcam
ab224921) at 1/1000 dilution (1% BSA, 0.1% Tween 20, PBS). Performed a second
washing with PBS for three times. DAPI was used to stain the cell nuclei.

**Flow cytometry**. In total, $1 \times 10^6$ Cells were digested and washed twice with PBS,
then staining with Annexin V Apoptosis Detection Kit (eBioscience 88-8007-74)
following the instructions. Briefly, cells were resuspended in 1X binding buffer
containing APC-conjugated Annexin V and incubated in dark for 10–15 min at
room temperature. After washing with 1X binding buffer, cells were resuspended in
1X binding buffer containing Propidium Iodide. Cells were filtered with 70 um
filter and analyzed by flow cytometry (LSRFortessa, BD Bioscience) within 4 h.
Data were analyzed using FlowJo software. Figure exemplifying the gating strategy
is provided in Supplementary Fig. 5.

**Biolayer interferometry (BLI) assay**. BLI assay was performed as previously
described with modifications[68]. Biotin-labeled RNAs were immobilized on strep-
tavidin biosensors by soaking the biosensors in 200 μl PBS containing 100 nM
RNA. To measure the association of PACT with full length *DARS-AS1*, biosensors
were incubated in 200 μl kinetic buffer (PBS supplied with 0.02% Tween 20)
containing 1 μM PACT for 200 s. This step was prolonged to 240 s when associated
with *DARS-AS1* fragments. The dissociation was measured in kinetic buffer
without PACT. All steps were performed in the OctetRed system (Pall ForteBio,
CA, USA) with 800 rpm shaking. Data were analyzed with Data Analysis 9. Binding
curves were aligned to Y-axis and the association-dissociation inter-step curve. $K_D$
values between PACT and RNAs binding were calculated using GraphPad Prime
8.0 software. Binding curves were fitted as nonlinear regression using association-
then-dissociation model. We constrained HotNM (100; means the concentration of
RNA we used in this assay was 100 nM) and Time0 (180 for full length DARS-AS1,
240 for RNA fragments; means the time at which dissociation was initiated) to
constant values. Other parameters were utilized the default values of the software.

**Mice**. SW620 cells transfected with *DARS-AS1* shRNA or control shRNA were
injected subcutaneously into BALB/c nude female mice ($2 \times 10^5$ cells per mouse,
$n = 10$, 6–8 weeks old). Tumors were observed 5 days after injection. Tumor
volumes were measured each 2 days. After 30 days, mice were sacrificed and
tumors were excised. The size and weight of tumors were measured. All studies
were performed in Laboratory Animal Research Center of Tsinghua University
under relative guidelines and approved by the Animal Care and Use Committee of
Tsinghua University.

**Data collection**. The FPKM values of *DARS-AS1* in TCGA normal and tumor samples and the clinical information of the patients were downloaded from UCSC Xena (http://xena.ucsc.edu).

**Kaplan–Meier survival analysis**. Patients were ranked based on the RNA expression level of *DARS-AS1*. The top 50% patients were defined as "high *DARS-AS1*" and the bottom 50% were defined as "low *DARS-AS1*". The survival curves were generated with the Kaplan–Meier method. The statistical significance was analyzed using log rank test.

**Statistics and reproducibility**. Results are presented as means ± standard deviation in triplicate experiments. Two-tailed Student's $t$ test was used to analyze the differences between groups. The level of significance was set at $*p < 0.05$, $**p < 0.01$, $***p < 0.001$, $****p < 0.0001$, ns, not significant. For xenograft model, instead, means ± standard error was used ($n = 10$).

**Reporting summary**. Further information on research design is available in the Nature Research Reporting Summary linked to this article.

## Data availability

All data reported in this study are available within this article and its Supplementary Information files. Uncropped images of blots are provided in Supplementary Information. Addgene IDs for plasmids are included in "Methods". Source data used to generate graphs and charts are provided in Supplementary Data 2 and 3. Results of the CRISPRi screening are available at the Gene Expression Omnibus (GEO) with the accession number of GSE197980.

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

## Acknowledgements

We thank Dr. Haiteng Deng, Yan Liu in Center of Biomedical Analysis, Tsinghua University, for their support with mass spectrum and flow cytometry analysis. We thank Daqing Jiang, Yang Cao for suggestions. This work was supported by grants from the National Natural Science Foundation of China (82172723, 31371314, 81673460), Science & Technology Department of Sichuan Province (2021ZYD0079) and Sichuan Youth Science and Technology Innovation Research Team of Experimental Formulology (2020JDTD0022).

## Author contributions

D.W., L.Y. and K.L. designed the research. L.Y. performed most of the wet-lab experiments, L.Z. helped with the mouse experiment, L.Z. and H.W. helped with RIP experiment, with the participation of S.T., L.H. and S.Z., K.L. performed most of the bioinformatics analysis. L.Y. and K.L. wrote the manuscript with the help from D.W., G.Z. and Z.J.L., all authors reviewed and edited the manuscript.

## Competing interests

The authors declare no competing interests.
