## [Peer Review File · Communications Biology]

Reviewers' comments:

Reviewer #1 (Remarks to the Author):

Yang et al. started with large-scale CRISPRi screening of lncRNAs functioning in cancer cell proliferation, and identified DARS-AS1 as a strong candidate in enhancing cancer proliferation. The authors validated this function through knocking down DARS-AS1 in multiple cancer cell lines. More importantly, with series of experiments, the authors revealed the potential molecular mechanism: DARS-AS1 directly binds to the activator domain of PACT, blocks PKR activation, and inhibits eIF2a phosphorylation and apoptotic cell death. The authors finally showed that this lncRNA is broadly expressed across multiple cancers and its high expression correlated with poor prognosis.

Overall, the data in this study are comprehensive; experimental design and results are conclusive; logic and writing are good. The finding of DARS-AS1/PACT/PKR axis should be of interest to others in the community.

Minor comments:

1. The authors mapped the A3 region of DARS-AS1 lncRNA as important interaction site with PACT. What is the unique feature of A3 compared with A1 and A2, in terms of RNA sequence and RNA secondary structure?
2. In Fig1B, NC need description. The Fold-Change should be labeled with which samples are compared.
3. The figure quality of Fig2C and F are not good, and difficult to notice the differences.
4. In Figure 4D, three shRNAs seem to have large variations and need clarifications.

Reviewer #2 (Remarks to the Author):

In this manuscript, Yang et al. reported a comprehensive study of an oncogenic long non-coding RNA, DARS-AS1. The authors have applied CRISPRi technology and high-throughput screening to discover new functional lncRNA genes in tumor growth. They confirmed their screening results with different in vitro and in vivo assays to show that loss-function of DARS-AS1 leads to reduced tumor growth and sphere formation. They also elucidated that DARS-AS1 functions through regulating PACT-PKR signaling pathway during the stress response of tumor cells. Their further studies showed that DARS-AS1 blocked the interaction between PACT and PKR through directly binding to PACT. The authors further analyzed the clinical data and identified a negative correlation between the expression level of DARS-AS1 with patient prognosis, indicating the potential clinical application of this lncRNA as new biomarkers.

This paper will bring new insights to our understanding of how long non-coding RNAs facilitate cancer cells' survival in stressful microenvironment. The authors integrated cellular and animal experiments, biochemistry assays, as well as bioinformatics analyses to supported their conclusions. The manuscript is well organized. Overall, this paper is of high quality and it potentially deserves the publication in Communications Biology. However, a number of concerns need to be addressed before the publication.

1. The authors have performed both knockdown and overexpression assays to validate the function of DARS-AS1 in several cell lines. The manuscript would benefit from more discussion of the reasons why they have picked these cell lines. It would be interesting to overexpress DARS-AS1 in a cell line with low expression of DARS-AS1 to see if the opposite phenotype is observed relative to DARS-AS1

knockdown in a high expressing cell line.

2.The authors have elucidated how DARS-AS1 regulates cellular stress responses by interrupt PACT functioning. It will more informative to add additional assays to check what is the effect of knockdown PACT alone on cancer cell growth.

3.In Figure 5F, the authors have proposed a model of DARS-AS1 in regulating PACT-PKR pathway. However, the schematic is kind of unclear. The author should make appropriate revisions to let the readers know better their findings.

Reviewer #3 (Remarks to the Author):

Summary:

In this study titled: "Long non-coding RNA DARS-AS1 promotes tumor progression by directly suppressing PACT-mediated cellular stress", Yang and Lin et al. investigated roles of lncRNAs in promoting cancer growth. The authors first used global CRISPRi screening of >900 lncRNAs and identified top candidates of cancer expressed lncRNAs that can alter cell growth. Among these, they found DARS-AS1 as a key lncRNA in several cancer cell lines. They used both in vitro and in vivo methods that validated the role of this lncRNA in promoting cell/tumor growth. They further conducted biochemical studies of lncRNA DARS-AS1 binding proteins, and identified an interesting pathway that DARS-AS1 mediated PACT/PKR regulation to promote cancer survival under cell stress conditions.

The overall design of the work is robust, the logic is clear, the results were well-presented and are in support of the central conclusions. The authors extensively conducted biochemical characterization of DARS-AS1 interacting with PACT, which in turn supports that this process inhibits PKR activation. Overall, this work will of good interests to the field of lncRNA biology and is new in terms of lncRNA regulation of PKR activation during cell stress response. I overall support the publication of this paper in Communications Biology, but there are problems in a few places that need to be addressed.

Major Comments:

- The CRISPRi gRNA library for lncRNAs is nice and useful. But I cannot find the introduction of the design strategy of the gRNAs (e.g., did the gRNAs targeting lncRNA promoter? TSSs or how are they designed?), the sequences and genomic locations of these gRNAs, or what are the non-targeting control gRNAs used in for this library or for screening. These should be made available in terms of the gRNA sequences/locations, and I would also encourage the authors to make available the gRNA library plasmid mix for public sharing (e.g. addgene).

- As the central target in this paper, gRNAs and shRNA sequences targeting DARS-AS1 and the relative positions of such gRNAs/shRNAs can be illustrated as a supplementary figure (e.g., in Fig.S2A, positions of shRNA/gRNAs relative to the DARS/DAR-AS1 can be shown as BLAT results in UCSC browser, which will make it easier to interpret Fig.S2B).

- Sequencing depth and replicate consistency for the CRISPRi screening shall be included in supplemental tables.

- The CRISPRi screening results cannot be found – the authors should deposit them in a public database such as GEO.

- While the functions of DARS-AS1 were tested quite well in this paper by several methods, it is helpful to know the absolute expression level of the lncRNA. Can the authors provide its expression level relative to GAPDH or other internal controls, and can estimate the copy number of this lncRNA in

the few cancer cell lines at least in Sw620 or A549? The expression levels of the lncRNA will make it more interpretable than how it impacts PACT/PKR regulation in the cytosol.

- In a few places, the author mentioned Kd values between PACT and lncRNA binding (e.g. page 9, line 22), but how this was calculated was not described in methods.

- Fig.4B, did knockdown of DARS-AS1 increase PACT-PKR interaction?

-Fig.4G,H, I am not following the authors very well on these data. The mutA of PACT binds DARS-AS1 poorly (Fig.4G), but its overexpression somehow can rescue the inhibition of DARS-AS1 in Fig.4H. In cells with DARS-AS1 inhibition (by shRNA in Fig.4H), PACT activates PKR more, but how come the mutA of PACT can further increase cell growth? It can compete with WT PACT to bind PKR? Also, how about mutD in cell growth assays as in Fig.4G,H? The authors shall clearly explain these plots in Fig.4G,H, or may consider to shorten/move some of these to supplementary figures (or remove entirely from the paper) as I found them peripheral to the central conclusion, and are not straightforward to interpret.

-Fig.5F is a helpful diagram. Have the authors tested some stress conditions, and does DARS-AS1 level get increased by some stress in cancer cells, for example, by thapsigargin?

Minor Comments:

- page 13, thapsgargin should be thapsigargin? Can the authors briefly introduce what is this, and what is the purpose of using this here? This is for the readers' information.

- Fig. 1E-M, these are comparison of values in two different populations of samples, so Mann Whitney U tests may be more suitable for statistical tests.

- Fig. 3H, the in vitro RIP is interesting that the delta-d2 (the loss of the second dsRNA binding domain) of PACT actually binds the lncRNA better than the WT protein, how the authors interpret this data?

- Fig. 3H, the western blot for the PACT protein in the right lower corner can be shown for a bit larger area, the current blot was cropped too small.

- Fig. 4F, can the authors quantify the changes in Western blots for p-PKR and for PACT reduction?

POINT-BY-POINT RESPONSES TO THE REVIEWERS

Summary of our responses:

We greatly appreciate the positive comments from all three reviewers such as “...the data in this study are comprehensive; experimental design and results are conclusive; logic and writing are good. The finding of DARS-AS1/PACT/PKR axis should be of interest to others in the community.” and “...This paper will bring new insights to our understanding of how long non-coding RNAs facilitate cancer cells’ survival in stressful microenvironment.... this paper is of high quality and it potentially deserves the publication in Communications Biology.” and “The overall design of the work is robust, the logic is clear, the results were well-presented... this work will of good interests to the field of lncRNA biology and is new in terms of lncRNA regulation of PKR activation during cell stress response.”.

We greatly appreciate the reviewers for their time and effort on our manuscript. We performed extra experiments and carefully revised the manuscript to address every single question and comment raised by reviewers (All the changes are marked in red in revised manuscript). The detailed summary of our revisions is listed below:

1. Revision of figures and tables:

- (1) **Figure 1B** was revised by labeling with which samples are compared to calculate the Fold-Change;
- (2) **Figure 2C and 2B** were revised by adding amplified views;
- (3) **Figure S2A** was revised by labeling the positions of sgRNAs and shRNAs on the schematic diagram of gene loci;
- (4) **New Figure S3D** and **new Figure S3E** were added into the **revised Supplementary Figure 3** to address the effect of knockdown PACT alone on cancer cell growth. The order of the figures has been changed (**old Figure S3D, S3E, S3F** were changed to **Figure S3F, S3G, S3H**, respectively);
- (5) **Figure 4D** was revised by quantifying the change in Western blots for p-PKR and p-eIF2 α ;

- (6) **Figure 4F** was revised by quantifying the changes in Western blots for p-PKR and for PACT reduction;
- (7) **Old Figure 4G** and **old Figure 4H** were moved to **revised Supplementary Figure S4D** and **S4E**. The order of the figures has been changed (**old Figure 4I, 4J, 4K** were changed to **Figure 4G, 4H, 4I**, respectively);
- (8) **New Figure S4C** were added into the **revised Supplementary Figure 4** to address the influence on the expression of DARS-AS1 by thapsigargin. The order of the figures has been changed (**old Figure S4C, S4D, S4E** were changed to **Figure S4F, S4G, S4H**, respectively);
- (9) **Figure 5F** was revised to better illustrate the function of DARS-AS1 in regulating PACT-PKR pathway;
- (10) **Table S1** was revised by adding the sequences of sgRNAs targeting DARS-AS1;
- (11) **New Supplementary Table 4** were added to address the expression level of DARS-AS1 in cell lines;
- (12) **New Supplementary Table 5** and **new Supplementary Table 6** were added to better annotate the CRISPRi sgRNA library and the results of screening;
- (13) **Figure S1C** was removed because the source data was deprecated by the data provider¹ (<http://research-pub.gene.com/KlijnEtAl2014/>).

2. Revision of manuscript:

- (1) More explanation about why we picked these lines to study was added in revised manuscript (**Page 9, line 10 to 12 & Page 11, line 11 to 13**);
- (2) The results of effect on SW620 cell growth by knocking down PACT were added in revised manuscript (**Page 13, line 21 to 22 & Page 14, line 1 to 3**);
- (3) More detailed explanation about the function of thapsigargin and the purpose we using it in this study were added in revised manuscript (**Page 18, line 10 to 16**);
- (4) The results of influence on DARS-AS1 expression upon thapsigargin treatment were added in revised manuscript (**Page 19, line 2 to 3**);

- (5) The method to introduce the design of sgRNA library was added in revised manuscript (**Page 28, line 6 to 8**);
- (6) The method of how to calculate K_D in BLI assay was added in revised manuscript (**Page 35, line 3 to 9**);
- (7) The title of section “Statistical analysis” in Method was revised as “Statistics and Reproducibility” and added the sample sizes and number of replicates (**Page 36, line 7**).

3. The CRISPRi screening results have been deposited in GEO.

We added a separate section titled “Data Availability” to provide the accession number (**Page 36, line 14 to 16**).

Reviewer #1 (Remarks to the Author):

Yang et al. started with large-scale CRISPRi screening of lncRNAs functioning in cancer cell proliferation, and identified DARS-AS1 as a strong candidate in enhancing cancer proliferation. The authors validated this function through knocking down DARS-AS1 in multiple cancer cell lines. More importantly, with series of experiments, the authors revealed the potential molecular mechanism: DARS-AS1 directly binds to the activator domain of PACT, blocks PKR activation, and inhibits eIF2a phosphorylation and apoptotic cell death. The authors finally showed that this lncRNA is broadly expressed across multiple cancers and its high expression correlated with poor prognosis.

Overall, the data in this study are comprehensive; experimental design and results are conclusive; logic and writing are good. The finding of DARS-AS1/PACT/PKR axis should be of interest to others in the community.

We greatly appreciate the Reviewer for the positive comments and the recognition of the significance of our study.

Minor comments:

1. The authors mapped the A3 region of DARS-AS1 lncRNA as important interaction site with PACT. What is the unique feature of A3 compared with A1 and A2, in terms of RNA sequence and RNA secondary structure?

We thank the Reviewer for the insightful question. Our data showed that PACT interacts with the A3 region of DARS-AS1, which is about ~400nt in length (see details below). We checked the RNA sequences and RNA secondary structure using RNAfold web server (<http://rna.tbi.univie.ac.at/cgi-bin/RNAWebSuite/RNAfold.cgi>). Unfortunately, we didn't observe significant and unique feature of A3 (see **Response Letter Fig. 1 below**). All three fragments form several stems and loops with similar size

and distribution pattern. Usually, the interaction site/motif of a RNA that interacts with a protein is about 20nt². Therefore, more detailed work is required to explore the exact motif and its secondary structure within A3. However, we think these results are beyond the scope of this manuscript, and we would like to explore them in the next project.

RNA sequence of A3:

```
GACTGGTCTCTTTTCTCCAATGTGTCCCTAACAGAGTGGTGAGGCTGGCTC
TTCCCACCAGTACAGGAAGATCATTCTTAAAAGAAATAGCCATATGGCT
TATAAGGGACTATATCTTAAGACTATGTGTCTTCTGCAGTTCTTTGCATAG
AGTCAACGTTGAAGGACAAGCAACTTCAAATTTCCAAGCAACAAAATTAT
GTTTCGGGTGGTTTTCTGAAAGCGAGATCTTAAGAACCACCTGTATCAGAA
TCCCATGAAGAGCTTGTTTAGTGTGCAAATTTACAGATTGTCCCCCCTCAG
AAATGCTCATTAGGAGTTCCTGTCCTTTGCGGGGACATGGATGAAGCTG
GAAACCATCATTCTCAGCAAACCTAACACAGG
```

minimum free energy

Response Letter Fig. 1. Secondary structure of A1, A2 and A3 region of DARS-AS1 predicted using RNAfold web server (<http://rna.tbi.univie.ac.at/cgi-bin/RNAWebSuite/RNAfold.cgi>).

2. In Fig1B, NC need description. The Fold-Change should be labeled with which samples are compared.

We thank the Reviewer for this suggestion. In Fig. 1B, NC stands for non-targeting sgRNAs in the sgRNA library and showed as black dots. Fold change of each sgRNA

was calculated by compared its normalized reads in samples of day17 with day 0. Following the Reviewer's suggestion, we have revised Fig. 1B and the legend of Fig. 1B. (Revised manuscript: Fig. 1B; see Response Letter Fig. 2 below).

Response Letter Fig. 2. (Revised Fig. 1B) Enrichment of sgRNAs after screening. The horizontal dotted lines represent $\log_2(\text{Fold change}) = \pm 0.58$. The vertical dotted line represents $P\text{-value} = 0.05$. Black dots are non-targeting sgRNAs (marked as NC). Red dots are sgRNAs targeting DARS-AS1. Blue dots are sgRNAs targeting LINC00205, a previously reported oncogenic lncRNA. **Fold change = (normalized reads day17) / (normalized reads day0)**.

3. The figure quality of Fig. 2C and F are not good, and difficult to notice the differences. Following the Reviewer's suggestion, we added amplified views of Fig. 2C and 2F in the revised manuscript (Revised manuscript: Fig. 2C and 2F; see Response Letter Fig. 3 below). It's observed that knocking down DARS-AS1 reduced the number of cellular colonies, while over-expression of DARS-AS1 increased. Combined with the MTS assay (Fig. 2A) and sphere formation assay (Fig. 2B), these results indicate that DARS-AS1 promotes cancer cell growth *in vitro*.

C

Colony formation assay

Response Letter Fig. 3. (Revised Fig. 2C and 2F) DARS-AS1 knockdown (C) inhibits the ability of colony formation in SW620 cell line, while overexpression of DARS-AS1 (F) increases colony formation.

4. In Figure 4D, three shRNAs seem to have large variations and need clarifications.

We thank the Reviewer for this question. In this experiment, we have designed three shRNA sequence targeting different regions of DARS-AS1, which may influence its function with varying degree. For better comparison, we quantified the change of p-PKR and p-eIF2 α compared to control (nc) after being normalized by β -tubulin of each sample (Revised Fig. 4D; see Response Letter Fig. 4 below). Although there are some variations, the level of p-PKR and p-eIF2 α are still notably increased by shRNAs targeting DARS-AS1, indicating that DARS-AS1 clearly blocks the phosphorylation of PKR and eIF2 α in cancer cell.

Response Letter Fig. 4. (Revised Fig. 4D) DARS-AS1-shRNAs treatment followed by serum starvation increases the level of p-PKR and p-eIF2 α in SW620 and HCT116 cells.

Reviewer #2 (Remarks to the Author):

In this manuscript, Yang et al. reported a comprehensive study of an oncogenic long non-coding RNA, DARS-AS1. The authors have applied CRISPRi technology and high-throughput screening to discover new functional lncRNA genes in tumor growth. They confirmed their screening results with different in vitro and in vivo assays to show that loss-function of DARS-AS1 leads to reduced tumor growth and sphere formation. They also elucidated that DARS-AS1 functions through regulating PACT-PKR signaling pathway during the stress response of tumor cells. Their further studies showed that DARS-AS1 blocked the interaction between PACT and PKR through directly binding to PACT. The authors further analyzed the clinical data and identified a negative correlation between the expression level of DARS-AS1 with patient prognosis, indicating the potential clinical application of this lncRNA as new biomarkers.

This paper will bring new insights to our understanding of how long non-coding RNAs facilitate cancer cells' survival in stressful microenvironment. The authors integrated cellular and animal experiments, biochemistry assays, as well as bioinformatics analyses to supported their conclusions. The manuscript is well organized. Overall, this paper is of high quality and it potentially deserves the publication in Communications Biology. However, a number of concerns need to be addressed before the publication.

We greatly appreciate the Reviewer for the positive comments and the recognition of the importance of our study.

1. The authors have performed both knockdown and overexpression assays to validate the function of DARS-AS1 in several cell lines. The manuscript would benefit from more discussion of the reasons why they have picked these cell lines. It would be interesting to overexpress DARS-AS1 in a cell line with low expression of DARS-AS1

to see if the opposite phenotype is observed relative to DARS-AS1 knockdown in a high expressing cell line.

We thank the Reviewer for this insightful comment. To validate the function of DARS-AS1 in multiple cell lines, we first checked the expression level of DARS-AS1 in some cell lines by RT-qPCR. We picked HCT116, MDA-MB-231 and HepG2 cell lines, with higher expression level of DARS-AS1, to do the knock-down experiment. Furthermore, following the suggestion from the Reviewer, we select A549 cell line with low expression of DARS-AS1 to do the overexpression experiment. As showed in Fig. 2 and Fig. S2, overexpressing DARS-AS1 in A549 cell promotes cell growth, while knockdown DARS-AS1 in HCT116, MDA-MB-231 and HepG2 cells inhibit cell growth. These data indicate DARS-AS1 promotes growth in multiple cancer cell lines. We also added words in revised manuscript to explain why we picked these cell lines to study (Page 9, line 10 to 12 & Page 11, line 11 to 13, “Beside SW620, we also used three other cell lines with higher expression level of DARS-AS1 to detect the knockdown efficiency and function of shRNAs” and “We used another lower DARS-AS1-expressed cell line, A549, to confirm this result. This cell proliferation enhancement by overexpression of DARS-AS1 was further observed in A549 cells”).

2.The authors have elucidated how DARS-AS1 regulates cellular stress responses by interrupt PACT functioning. It will more informative to add additional assays to check what is the effect of knockdown PACT alone on cancer cell growth.

Following the Reviewer’s suggestions, we knocked down PACT in SW620 cells to check the influence on cell growth by measuring the cell growth rate and the ability of colony formation. Our results show that PACT knockdown increases SW620 cell growth, as well as the numbers of cellular colonies. We have already showed overexpression of PACT decreases cellular growth (Fig. 3I). All these data indicates that PACT knockdown promotes cancer cell growth, which is opposite to DARS-AS1. The results were added in new Fig. S3D and S3E (see Response Letter Fig. 5 below) of the Revised Supplementary Information and the revised manuscript (Page 13, line 21

to 22 & Page 14, line 1 to 3, “We first detected the effect of knocking down PACT on cell growth. We noticed that, on the contrary with DARS-AS1, the relative cell growth was 1.5 to 3 times faster when knocking down PACT (Figure S3D). The results of colony formation assay showed cells formed 2 to 3 fold more colonies after PACT shRNA treatment. (Figure S3E).”).

Response Letter Fig. 5. (Revised Fig. S3D, S3E) PACT knockdown increases cell growth (D) and colony formation (E) in SW620 cells.

3. In Figure 5F, the authors have proposed a model of DARS-AS1 in regulating PACT-PKR pathway. However, the schematic is kind of unclear. The author should make appropriate revisions to let the readers know better their findings.

We thank the Reviewer for the suggestions. We have revised Fig. 5F.

Response Letter Fig. 6. (Revised Fig. 5F) Proposed model of DARS-AS1 in regulating PACT-PKR pathway and tumor growth.

Reviewer #3 (Remarks to the Author):

Summary:

In this study titled: “Long non-coding RNA DARS-AS1 promotes tumor progression by directly suppressing PACT-mediated cellular stress”, Yang and Lin et al. investigated roles of lncRNAs in promoting cancer growth. The authors first used global CRISPRi screening of >900 lncRNAs and identified top candidates of cancer expressed lncRNAs that can alter cell growth. Among these, they found DARS-AS1 as a key lncRNA in several cancer cell lines. They used both in vitro and in vivo methods that validated the role of this lncRNA in promoting cell/tumor growth. They further conducted biochemical studies of lncRNA DARS-AS1 binding proteins, and identified an interesting pathway that DARS-AS1 mediated PACT/PKR regulation to promote cancer survival under cell stress conditions.

The overall design of the work is robust, the logic is clear, the results were well-presented and are in support of the central conclusions. The authors extensively conducted biochemical characterization of DARS-AS1 interacting with PACT, which in turn supports that this process inhibits PKR activation. Overall, this work will of good interests to the field of lncRNA biology and is new in terms of lncRNA regulation of PKR activation during cell stress response. I overall support the publication of this paper in Communications Biology, but there are problems in a few places that need to be addressed.

We greatly appreciate the Reviewer for the positive comments and the recognition of the importance of our study.

Major Comments:

1. The CRISPRi gRNA library for lncRNAs is nice and useful. But I cannot find the introduction of the design strategy of the gRNAs (e.g., did the gRNAs targeting lncRNA promoter? TSSs or how are they designed?), the sequences and genomic locations of these gRNAs, or what are the non-targeting control gRNAs used in for this library or for screening. These should be made available in terms of the gRNA sequences/locations, and I would also encourage the authors to make available the gRNA library plasmid mix for public sharing (e.g. addgene).

We thank the Reviewer for these insightful questions. We revised the method to introduce the design of sgRNA library (Page 28, line 6 to 8, “The sgRNAs were designed using a public tool named CRISPR-ERA³. We utilized the default parameters of this tool to design the sgRNAs and the algorithm computes sgRNA binding sites within a 3 kb region centered at TSS.”). The detailed information of these sgRNAs including 500 non-targeting control sgRNAs has been added in the **Supplementary Table. 5** of the revised manuscript. Due to intellectual property issues, we are in a process of communicating with our college before the plasmid library can be available on addgene. Once it is approved, we would like to share the plasmid library with the public for scientific research.

2. As the central target in this paper, gRNAs and shRNA sequences targeting DARS-AS1 and the relative positions of such gRNAs/shRNAs can be illustrated as a supplementary figure (e.g., in Fig.S2A, positions of shRNA/gRNAs relative to the DARS/DAR-AS1 can be shown as BLAT results in UCSC browser, which will make it easier to interpret Fig.S2B).

Following the Reviewer’s suggestions, we illustrated the positions of sgRNAs and shRNAs in **revised Fig. S2A** (see **Response Letter Fig. 7** below). The sequences of sgRNAs were added in **revised Table. S1**. sgRNAs are located close to the TSS shared by DARS-AS1 and DARS, so that can inhibit the expression of both genes. shRNAs

are located in the 3' end of DARS-AS1 gene, which only influence the mRNA of DARS-AS1. We thank the Reviewer's suggestions.

Response Letter Fig. 7. (Revised Fig. S2A) Schematic representation of DARS-AS1 and DARS genomic locus, and the positions of shRNAs and sgRNAs.

3. Sequencing depth and replicate consistency for the CRISPRi screening shall be included in supplemental tables.

We thank the Reviewer for the suggestions. The sequencing depth ranges from 526 to 652. The consistency (R-Squared) of each two samples ranges from 0.97 to 0.76. We added the details of sequencing depth and replicate consistency in the Supplementary Table. 6 (see Response Letter Table. 1 below) in the revised manuscript.

Sample	total reads	sequencing depth
SW620_day0_rep1	5124383	652
SW620_day0_rep2	4834548	615
SW620_day0_rep3	4298508	547
SW620_day0_rep4	4734102	603
SW620_day17_rep1	4783560	609
SW620_day17_rep2	4302854	548
SW620_day17_rep3	4134830	526
SW620_day17_rep4	4183347	533

Response Letter Table. 1 (Revised Supplementary Table. 6) The total reads and sequencing depth of screening (upper) and replicate consistency (R-Squared) of each two samples(below).

4. The CRISPRi screening results cannot be found – the authors should deposit them in a public database such as GEO.

We thank the Reviewer for this suggestion. The CRISPRi screening results have been deposited in GEO with the accession number of GSE197980. To review GEO accession GSE197980, please refer to (<https://www.ncbi.nlm.nih.gov/geo/query/acc.cgi?acc=GSE197980>).

5. While the functions of DARS-AS1 were tested quite well in this paper by several methods, it is helpful to know the absolute expression level of the lncRNA. Can the authors provide its expression level relative to GAPDH or other internal controls, and can estimate the copy number of this lncRNA in the few cancer cell lines at least in Sw620 or A549? The expression levels of the lncRNA will make it more interpretable that how it impacts PACT/PKR regulation in the cytosol.

We thank the Reviewer for the insightful question. Following the Reviewer’s suggestion, we checked the expression of DARS-AS1 in cell lines we used here by RT-qPCR using ACTB as internal control. We used similar amount of cDNA of each cell sample to make Ct values of ACTB about 14-16. The details of Ct values are listed below. SW620 and HCT116 are highly expressed DARS-AS1, with Ct values between

23.7~24.8. DARS-AS1 is moderately expressed in HepG2 and MDA-MB-231 (Ct: 26.1~26.8), while even lower in A549(Ct: 27.1~27.7), the cell line we used as lowly-expressed cell line here.

As cell lines are simplified models for cancer research, for better evaluating the expression of DARS-AS1, we analyzed its expression in tumor samples from TCGA database and compared with other functional lncRNAs. CCAT2 is highly expressed in colorectal cancer and promotes tumor growth⁴. In Colon adenocarcinoma (COAD), the mean FPKM (Fragments per kilobase per million mapped reads) of DARS-AS1(mean FPKM=0.16) is similar to CCAT2(mean FPKM=0.11), while the median FPKM of DARS-AS1(median FPKM=0.13) is higher than CCAT2(median FPKM=0.03). HOTAIR is a well-known oncogene-like lncRNA in multiple cancer types including COAD. The FPKM of DARS-AS1 and HOTAIR are comparable in both COAD or total tumor samples. These data indicate that the expression level of DARS-AS1 is sufficient to perform its function.

Beside us, some groups also reported DARS-AS1 promotes tumor progression in many other tumors including non-small cell lung cancer⁵, thyroid cancer⁶ and clear cell renal cell carcinoma⁷. These researches further prove that the expression level of DARS-AS1 is adequate for influencing cancer cell growth.

Response Letter Fig. 8. The expression level of DARS-AS1 in cell lines and tumor samples. (A (Revised Table. S4) and B).C_t values of DARS-AS1 and ACTB (internal control) in cell lines. (C).The relative expression level of DARS-AS1 compared with ACTB. $\log_2\text{FoldChange} = -(C_{tDARS-AS1} - C_{tACTB})$. (D and E).The FPKM (Fragments per kilobase per million mapped reads) of DARS-AS1 in all tumor samples (D) or COAD samples (E) from TCGA database.

6. In a few places, the author mentioned K_d values between PACT and lncRNA binding (e.g. page 9, line 22), but how this was calculated was not described in methods.

Following the Reviewer's suggestion, we described the methods of calculating K_D in the revised manuscript (Page 35, line 3 to 9, "K_D values between PACT and RNAs binding were calculated using GraphPad Prime 8.0 software. Binding curves were fitted as nonlinear regression using association-then-dissociation model. We constrained HotNM (100; means the concentration of RNA we used in this assay was 100 nM) and Time0 (180 for full length DARS-AS1, 240 for RNA fragments; means the time(second) at which dissociation was initiated) to constant values. Other parameters were utilized the default values of the software.").

7. Fig.4B, did knockdown of DARS-AS1 increase PACT-PKR interaction?

We thank the Reviewer for the questions. We performed extra co-immunoprecipitation (co-IP) assays to answer this question. SW620 cells were treated by shRNAs targeting DARS-AS1 or control shNC followed by serum starvation. Cell lysates were used for co-IP analysis with anti-PACT antibody. Results showed that more PKR were retrieved by anti-PACT antibody in cells treated by shDARS-AS1 (Response Letter Fig. 8), indicating knockdown of DARS-AS1 increases PACT-PKR interaction.

We have already done a series of assays to prove that DARS-AS1 interferes PACT-PKR interaction. First, we detected the *in situ* localization of PACT and PKR in cells by fluorescence confocal microscope and found that overexpression of DARS-AS1 reduces the colocalization of PACT and PKR (Fig. 4A). Then, we investigated PACT-PKR interaction in the cell by native co-immunoprecipitation assays and results showed that overexpression of DARS-AS1 blocks the PACT-PKR interaction (Fig. 4B). Further, we did *in vitro* protein binding assays with purified PACT, PKR and DARS-AS1. These assays excluded interference of other molecules in the cell and indicated that DARS-AS1 directly inhibits the association of PACT-PKR (Fig. 4C). All these results support our conclusion via multiple ways.

Response Letter Fig. 8. Knockdown of DARS-AS1 increased PACT-PKR interaction as showed in co-immunoprecipitation (co-IP) assays.

8. Fig.4G,H, I am not following the authors very well on these data. The mutA of PACT binds DARS-AS1 poorly (Fig.4G), but its overexpression somehow can rescue the inhibition of DARS-AS1 in Fig.4H. In cells with DARS-AS1 inhibition (by shRNA in

Fig.4H), PACT activates PKR more, but how come the mutA of PACT can further increase cell growth? It can compete with WT PACT to bind PKR? Also, how about mutD in cell growth assays as in Fig.4G,H? The authors shall clearly explain these plots in Fig.4G,H, or may consider to shorten/move some of these to supplementary figures (or remove entirely from the paper) as I found them peripheral to the central conclusion, and are not straightforward to interpret.

We thank the Reviewer for the question. MutA is the dominant inactive mutant of PACT, which still can bind to PKR but prevents activation of PKR in stressed cells⁸. Consistent with this report, we found mutA does not inhibit cell survival. Instead, overexpression of mutA slightly increases cell growth, possibly because the amount of heterogenous expressed mutA proteins are far more than WT PACT proteins, so that mutA competes with WT to bind PKR and abrogates PKR induced apoptosis. In control cells without expression of mutant PACT, shDARS-AS1 can inhibit cell growth at least partly through facilitating PACT-mediated PKR activation (as showed in original Fig. 4H, ctrl cells growth faster than ctrl+shDARS-AS1 cells). However, in cells overexpressing mutA, since DARS-AS1 poorly binds to PACT, shDARS-AS1 no longer influences cell growth (as showed in original Fig. 4H, the growth rate of PACTmutA cells is nearly with PACTmutA+shDARS-AS1 cells). These results indicated that (1) DARS-AS1 binds to D3 (the third domain of PACT) and (2) DARS-AS1 functions through PACT. Following the Reviewer's suggestions, we move the original Fig. 4G and Fig. 4H to supplementary figure (**revised Fig. S4D and S4E**). We thank the Reviewer again for this question.

9. Fig.5F is a helpful diagram. Have the authors tested some stress conditions, and does DARS-AS1 level get increased by some stress in cancer cells, for example, by thapsigargin?

We thank the Reviewer for this insightful question. Following the Reviewer's advice, we examined the expression of DARS-AS1 under treatment of different concentration

of thapsigargin (10nM or 50nM) or DMSO (as control) and added the results in revised manuscript (new Fig. S4C; Page 19, line 2 to 3, “Interestingly, thapsigargin induces DARS-AS1 expression in dose depended manner, which may imply the anti-stress function of DARS-AS1.”). Results showed that the expression of DARS-AS1 is upregulated under thapsigargin treatment in a dose depended manner (see Response Letter Fig. 9 below). It’s reported that DARS-AS1 is regulated by HIF-1⁹. We also noticed it is upregulated under hypoxia. These data indicate the expression of DARS-AS1 is inducible in stressed cells.

Response Letter Fig. 9. (Revised new fig. S4C). Thapsigargin treatment induces the expression of DARS-AS1 in a dose depended manner.

Minor Comments:

10- page 13, thapsgargin should be thapsigargin? Can the authors briefly introduce what is this, and what is the purpose of using this here? This is for the readers' information.

We thank the Reviewer for pointing out our spelling typo. Following the reviewer's suggestion, we give as more detail introduction of thapsigargin's molecular function and the purpose of using this here in the revised manuscript (Page 18, line 10 to 16, “Thapsigargin is an ER stressor, which causes Ca²⁺ release from ER. It is reported that thapsigargin treatment induces PACT expression and activation, which further interacts with and activates PKR, resulting in cell apoptosis via increasing eIF2 α

phosphorylation. Here, we used thapsigargin as a stimulator of PACT/PKR pathway to investigate whether DARS-AS1 can help cells overcome stress through suppressing PACT/PKR pathway.”).

11- Fig. 1E-M, these are comparison of values in two different populations of samples, so Mann Whitney U tests may be more suitable for statistical tests.

We thank the Reviewer for the valuable comment. In order to answer this question, we have carefully gone through literatures¹⁰⁻¹² to study the details of Mann Whitney U tests, unpaired two-sample Student's t test and paired two-sample Student's t test. The Mann-Whitney U test, which is also known as the Wilcoxon rank sum test, tests for differences between two groups on a single, ordinal variable with no specific distribution¹⁰. The Mann-Whitney U test is the nonparametric version of the unpaired two-sample Student's t test¹⁰. As the samples in our study are paired, neither the Mann-Whitney U test or the unpaired two-sample Student's t test is appropriate. Wilcoxon signed-rank test and paired two-sample Student's t test are suitable for the paired samples. And the Wilcoxon signed-rank test is the nonparametric version of the paired two-sample Student's t test. We think the parametric assumptions are met, so we have chosen the paired two-sample Student's t test to do the statistical tests. We thank the Reviewer again for this question.

12- Fig. 3H, the in vitro RIP is interesting that the delta-d2 (the loss of the second dsRNA binding domain) of PACT actually binds the lncRNA better than the WT protein, how the authors interpret this data?

We thank the Reviewer for this insightful question. PACT protein consists of two conserved double-stranded RNA binding domains (dsRBD) and a third domain (D3), with linker regions between each domain. Although no full-length structure of PACT was reported, researchers found these two dsRBDs are conserved and similar to dsRBDs of PKR, which may interact with each other. This intermolecular interaction

may form a complex structure and increase the steric hindrance around PACT, preventing the association of DARS-AS1. We predicted that, when the second dsRBD of PACT is deleted, there is less steric hindrance and a longer linker region between the first dsRBD and D3, which might make it easier for DARS-AS1 to bind with D3. Thus, the binding of delta-d2 of PACT to the DARS-AS1 might be stronger than the WT protein in our results.

13- Fig. 3H, the western blot for the PACT protein in the right lower corner can be shown for a bit larger area, the current blot was cropped too small.

Following the Reviewer's suggestion, we replaced the image of western blot with a larger area in revised Fig. 3H.

14- Fig. 4F, can the authors quantify the changes in Western blots for p-PKR and for PACT reduction?

We thank the Reviewer for this suggestion. Following this suggestion, we quantified the changes in revised Fig. 4F. The p-PKR or PACT in "IP" samples were first normalized by PKR or PACT in "input" samples. Then, each sample was compared with control group (without DARS-AS1 nor mock RNA). It's obvious that less PKR were phosphorylated in the presence of DARS-AS1, as well as less PACT binding to PKR, which demonstrated that DARS-AS1 inhibits PKR activation by blocking the interaction between PCAT and PKR.

F

Response Letter Fig. 10. (Revised new Fig. 4F). Thapsigargin treatment induces the expression of DARS-AS1 in a dose dependent manner.

Summary

We have thoroughly and carefully addressed every question or concern raised by the Reviewers through extensive experiments and analysis. We believe the soundness and comprehensiveness of our study has been significantly improved after the new results and discussions being added into the revised manuscript. We hope that this revised manuscript is now found to be suitable for publication in *Communications Biology*.

Reference

- 1 A comprehensive transcriptional portrait of human cancer cell lines. *Nature biotechnology* **33**, 306 (2015).
- 2 Zhao, W. *et al.* POSTAR3: an updated platform for exploring post-transcriptional regulation coordinated by RNA-binding proteins. *Nucleic acids research* **50**, D287-D294 (2022).
- 3 Liu, H. L. *et al.* CRISPR-ERA: a comprehensive design tool for CRISPR-mediated gene editing, repression and activation. *Bioinformatics* **31**, 3676-3678 (2015).
- 4 Ling, H. *et al.* CCAT2, a novel noncoding RNA mapping to 8q24, underlies metastatic progression and chromosomal instability in colon cancer. *Genome research* **23**, 1446-1461 (2013).
- 5 Liu, D., Liu, H., Jiang, Z., Chen, M. & Gao, S. Long non-coding RNA DARS-AS1 promotes tumorigenesis of non-small cell lung cancer via targeting miR-532-3p. *Minerva medica* (2019).
- 6 Zheng, W. *et al.* LncRNA DARS-AS1 regulates microRNA-129 to promote malignant progression of thyroid cancer. *European review for medical and pharmacological sciences* **23**, 10443-10452 (2019).
- 7 Jiao, M., Guo, H., Chen, Y., Li, L. & Zhang, L. DARS-AS1 promotes clear cell renal cell carcinoma by sequestering miR-194-5p to up-regulate DARS. *Biomed. Pharmacother.* **128** (2020).
- 8 Peters, G. A., Li, S. & Sen, G. C. Phosphorylation of specific serine residues in the PKR activation domain of PACT is essential for its ability to mediate apoptosis. *The Journal of biological chemistry* **281**, 35129-35136 (2006).
- 9 Mimura, I. *et al.* Novel lncRNA regulated by HIF-1 inhibits apoptotic cell death in the renal tubular epithelial cells under hypoxia. *Physiological Reports* **5** (2017).
- 10 Weiner, I. B. & Craighead, W. E. Mann-Whitney U Test. *The Corsini encyclopedia of psychology* (2010).
- 11 Mann, H. B. & Whitney, D. R. On a test of whether one of two random variables is stochastically larger than the other. *The annals of mathematical statistics*, 50-60 (1947).
- 12 Ozgur, C., Meek, G. & Dunning, K. Comparison of the t vs. Wilcoxon Signed-Rank Test for Likert Scale Data & Small Samples. *Journal of Modern Applied Statistical Methods* (2007).

REVIEWERS' COMMENTS:

Reviewer #1 (Remarks to the Author):

My questions have been addressed in the revision.

Reviewer #2 (Remarks to the Author):

Overall, I think the authors did a good job addressing all the concerns that I have raised during my first review and this manuscript has been greatly approved. I am satisfied with the revised manuscript and I support the publication.

Reviewer #3 (Remarks to the Author):

The authors have fully addressed my concerns in the original revision. I congratulate the authors for their work. This paper is now suitable to be published.